# Enveloped viruses distinct from HBV induce dissemination of hepatitis D virus in vivo

Jimena Perez-Vargas [1], Fouzia Amirache[1], Bertrand Boson[1], Chloé Mialon[1], Natalia Freitas[1], Camille Sureau[2], Floriane Fusil[1] & François-Loïc Cosset [1]

Hepatitis D virus (HDV) doesn't encode envelope proteins for packaging of its ribonucleo-protein (RNP) and typically relies on the surface glycoproteins (GPs) from hepatitis B virus (HBV) for virion assembly, envelopment and cellular transmission. HDV RNA genome can efficiently replicate in different tissues and species, raising the possibility that it evolved, and/or is still able to transmit, independently of HBV. Here we show that alternative, HBV-unrelated viruses can act as helper viruses for HDV. In vitro, envelope GPs from several virus genera, including vesiculovirus, flavivirus and hepacivirus, can package HDV RNPs, allowing efficient egress of HDV particles in the extracellular milieu of co-infected cells and sub-sequent entry into cells expressing the relevant receptors. Furthermore, HCV can propagate HDV infection in the liver of co-infected humanized mice for several months. Further work is necessary to evaluate whether HDV is currently transmitted by HBV-unrelated viruses in humans.

[1] CIRI—Centre International de Recherche en Infectiologie, Univ Lyon, Université Claude Bernard Lyon 1, Inserm, U1111, CNRS, UMR5308, ENS Lyon, 46 allée d'Italie, F-69007 Lyon, France. [2] Molecular Virology laboratory, Institut National de la Transfusion Sanguine (INTS), CNRS Inserm U1134, 6 rue Alexandre Cabanel, F-75739 Paris, France. Correspondence and requests for materials should be addressed to F.-L.C. (email: flcosset@ens-lyon.fr)

Hepatitis D virus (HDV) was discovered 40 years ago in the liver of individuals chronically infected with hepatitis B virus (HBV), a liver-specific pathogen present in ca. 250 million people. The HDV virion released in the extracellular milieu is an enveloped particle with an average diameter of 36 nm. It consists of a cell-derived lipid envelope harboring HBV surface proteins and coating an inner ribonucleoprotein (RNP)[1–4], which is composed of a multimer of ca. 70 copies of the HDV-encoded delta antigen (HDAg) protein[5,6] that is associated to one copy of the small circular single-strand HDV RNA exhibiting self-annealing properties, conferring its rod-like conformation[6,7]. Although HDAg was initially considered as a novel HBV antigen[8], it was later shown to be associated with a small RNA as a transmissible and defective agent that uses the HBV envelope glycoproteins (GP) for its propagation, hence reflecting its nature of an obligate satellite of HBV. Indeed, HDV particles appear not to require specific cellular functions to promote egress of its RNP and to only rely on the budding mechanism provided by HBV envelope GPs, which hence offers the exclusive HBV contribution to the HDV life cycle. Their ensuing envelopment subsequently allows targeting and entry of HDV particles to human hepatocytes via mechanisms that depend on the same host entry factors than those used by HBV itself, i.e., through low-affinity attachment to cell surface heparan sulfate proteoglycans (HSPGs)[9,10] and subsequent high-affinity engagement to the sodium taurocholate cotransporting polypeptide (NTCP)[11,12].

Noteworthy, the origin of HDV is currently unknown. The characterization of the HDAg-associated RNA, the HDV genome, revealed that it is unique among animal viruses and that it shares some properties with some plant agents called viroids[13,14]. Indeed, the replication of its RNA involves the HDAg-mediated subversion of cellular RNA polymerase(s), such as Pol-II. Both genomic HDV RNA and antigenomic RNA (its replication intermediate) strands include ribozyme autocatalytic, self-cleaving elements. Interestingly, cells from highly divergent organisms express several HDV-like cellular ribozymes that play a role in many biological pathways[15,16]. This has raised the possibility that HDV RNA originated from the cell transcriptome itself, in agreement with the finding that circular RNA species are abundant in cells[17]. Therefore, one possibility could be that the HDV RNA has emerged in HBV-infected hepatocytes subsequent to evolution of cellular circular RNA forms becoming autonomously replicative[18] and for which the ribozyme and HDAg-coding RNA sequences may have arisen from the human transcriptome[19,20]. Accordingly, that HBV, a strictly liver-tropic human pathogen, only provides RNP envelopment and transmission functions would therefore explain why HDV has been exclusively detected in the liver of HBV-infected patients. Alternatively, that HDV RNA can self-replicate in a much wider variety of cell types and species[21–23] raises the theoretical possibility that it may be transmitted through unorthodox means. Furthermore, viruses closely related to HDV have been detected in non-human species in the absence of any hepadnavirus[24,25]. Also, primary Sjögren's syndrome patients were reported to present HDV antigen and RNA in salivary glands in the absence of HBsAg or HBV antibodies[26].

Here, aiming to explore scenarios concerning the origin of HDV, we investigate the possibility that other, HBV-unrelated viruses could provide helper envelopment, budding, and entry functions. Our results indicate that HDV RNPs may exploit assembly functions provided by viruses from several alternative genera and families, including vesiculovirus, flavivirus, and hepacivirus, among other enveloped viruses. This compatibility allows efficient egress in the extracellular milieu of co-infected cells of HDV particles that appear to be infectious. This leads to their subsequent entry into different cell types expressing the receptors targeted by the GPs of either virus genus and dissemination of HDV genome in vivo in experimentally infected humanized mice.

## Results

**HDV particle assembly with vesiculovirus and hepacivirus GPs.** HDV particles were produced in Huh-7 cells by co-transfecting two plasmids, a first one providing the envelope GPs from HBV vs. alternative enveloped viruses, i.e., vesicular stomatitis virus (VSV) G protein and hepatitis C virus (HCV) E1E2 proteins, and a second one initiating the replication of the HDV RNA genome (pSVLD3[27]) that encodes the HDV delta protein (HDAg). As controls, we co-transfected with pSVLD3 an "empty" plasmid that does not encode GPs ("No GP") for assessing specific production, release, and infectivity of HDV particles. At 3, 6, and 9 days post transfection, the production of HDV particles was determined by quantifying by RT-qPCR of the HDV RNAs from the supernatants. While HDV RNAs accumulated at high levels in producer cells for all transfection conditions, reflecting its self-replication independently of GP expression, it was readily detected in the cell supernatants when the HBV GPs were co-expressed in transfected cells (Fig. 1a) with an over 4-log fold increase after day 3, in agreement with previous reports[28,29]. In contrast, no significant HDV RNA secretion could be detected over the RT-qPCR threshold levels when pSVLD3 was transfected without GP ("No GP" control), confirming that HDV RNA release from cells requires co-expression of HBV GPs[29].

Strikingly, we found that the release of HDV RNAs could also be induced by envelope GPs from alternative viruses, as suggested by progressively increased secretion of extracellular HDV RNAs over time post transfection (Fig. 1a). Specifically, at day 9 post transfection, we detected high levels of HDV RNA in the supernatants of cells co-expressing these GPs and HDV RNAs, by up to $10^9$ GE/mL for VSV-G GP-expressing cells, i.e., ca. sixfold higher than for HBV GPs, and by ca. $5 \times 10^7$ GE/mL for HCV-E1E2 GP-transfected cells (Fig. 1a). We confirmed that these extracellular RT-qPCR signals reflected bona fide HDV RNAs, as shown by strand-specific RT-PCR experiments that detected genomic HDV RNA at the expected size of 1.7 kb (Fig. 1b) and by northern blot experiments performed on pellets of ultracentrifuged supernatants from producer cells that revealed full-length HDV RNAs (Fig. 1c). Then, using a strand-specific RT-qPCR assay[30,31] that specifically quantifies either genomic (gRNA) or antigenomic (agRNA) HDV RNAs (Supplementary Fig. 1f), we found a strong enrichment of HDV gRNA in the supernatants of cells transfected by pSVLD3 and either of these GP expression plasmids, as compared with lysates of producer cells (Supplementary Fig. 1a–c). The HDV gRNAs accounted for over 99% of HDV RNAs detected in the supernatants of these cells (Supplementary Fig. 1d–e), suggesting that VSV and HCV GPs induced extracellular release of genomic HDV RNA in a manner similar to HBV GPs.

Next, we sought to investigate the biochemical form of these extracellular genomic HDV RNAs. As shown in Fig. 1e, we found that immunoprecipitation of producer cell supernatants with antibodies against HBV, VSV, or HCV GPs could co-precipitate HDV RNAs in a GP-specific manner, which suggested that the latter are in the form of GP-associated RNPs. In agreement with this possibility, when we immunoblotted the pellets of ultracentrifuged producer cell supernatants with HDAg antibodies (Fig. 1d), we found similar levels and ratios of L- and S-HDAg for particles generated with HCV and VSV GPs as compared with "normal" HDV particles produced with HBV GPs. This suggested that the detected genomic HDV RNAs (Fig. 1a–c) are

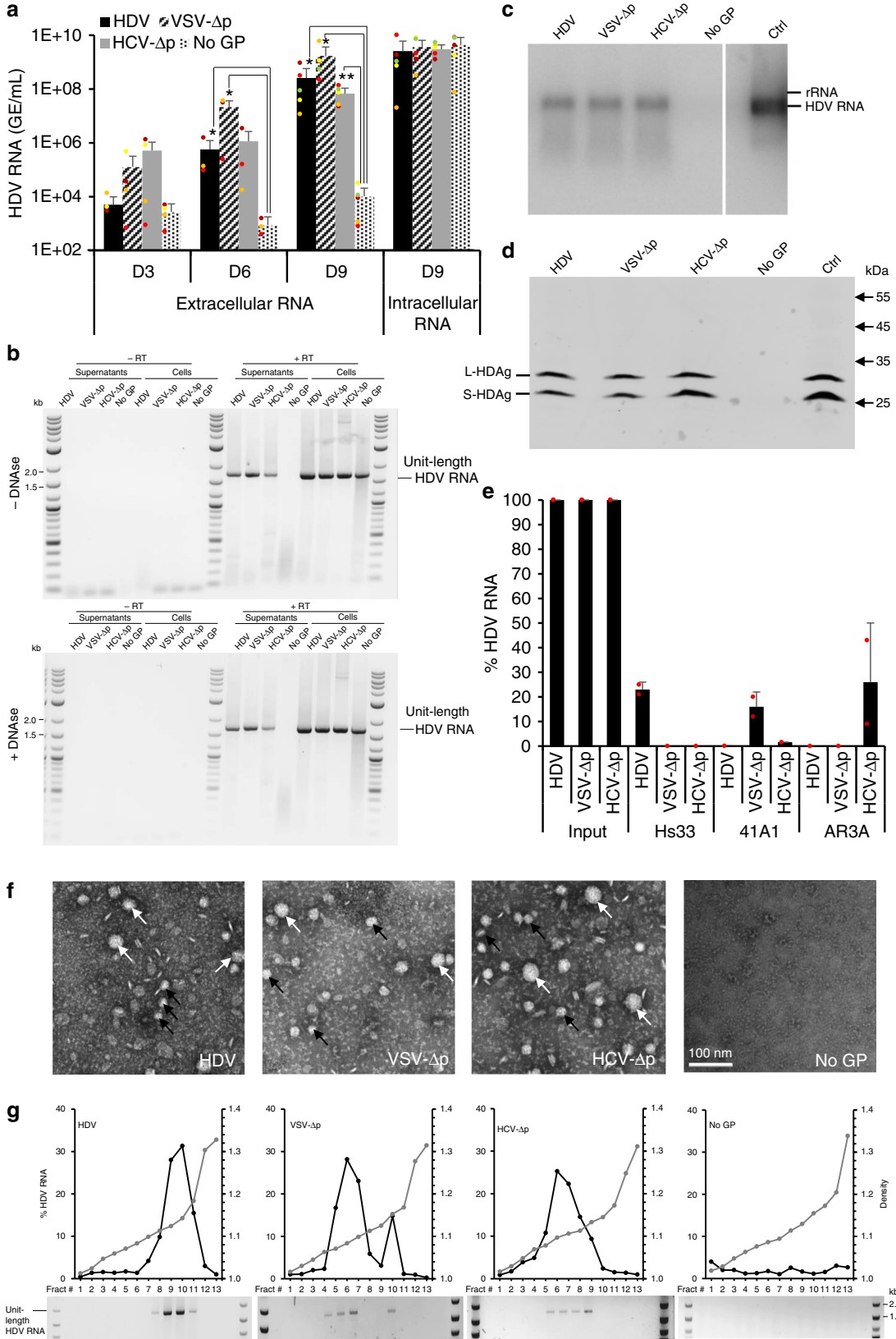

incorporated as RNPs exhibiting wild-type properties. Of note, co-expression of HDV RNPs with HCV and VSV GPs did not induce higher cytotoxicity levels than those detected in cells producing "normal" HDV particles or in non-transfected cells (Supplementary Fig. 2), suggesting genuine processes of envelopment and production of these novel HDV particles. Altogether,

these results indicated that HDV can be enveloped by different types of viral surface glycoproteins, which induces secretion of HDV RNPs in the extracellular milieu.

To further characterize the non-HBV-induced HDV particles (Δp) coated with VSV-G or HCV-E1E2 envelope GPs, hereafter designated VSV-Δp and HCV-Δp, we incubated the supernatants

**Fig. 1** Secretion of HDV particles is induced by surface glycoproteins from varied enveloped viruses. Huh-7 cells were co-transfected with pSVLD3 plasmid coding for HDV RNPs and plasmids coding for HBV, VSV, or HCV surface glycoproteins (GP), resulting in "HDV", "VSV-Δp", and "HCV-Δp" samples, respectively. As control, pSVLD3 was co-transfected with an empty plasmid ("No GP" samples). **a** At day 3, 6, or 9, extracellular HDV RNAs were quantified from cell supernatants by RT-qPCR. Intracellular HDV RNAs were quantified from cell lysates at day 9 post transfection. HDV RNA levels in GE (genome equivalent) are expressed as means ($n = 5$ independent experiments) per ml of cell supernatants for extracellular RNAs or, for intracellular RNAs, per mL of cell lysates containing $10^6$ cells. **b** RNAs extracted from lysates and supernatants of transfected cells treated with RNAse-free DNAse, or not treated (–DNAse), were reverse-transcribed using a antigenomic primer that detects HDV RNAs and then PCR-amplified with HDV-specific primers to reveal unit-length HDV genomic RNAs. As control, reverse transcriptase was omitted during processing of the samples (–RT). **c, d** In total, $2 \times 10^7$ HDV GEs from pellets retrieved after ultracentrifugation of cell supernatants on 30% sucrose cushions were analyzed by northern blot using a HDV-specific probe (**c**) or by western blot using an HDAg antibody (**d**). Control HDV RNAs ($5 \times 10^7$ GE) (**c**) or HDAg from cell lysates (**d**) were loaded on the same gels (Ctrl). **e** Pelleted cell supernatants containing $10^9$ HDV GEs ("Input") immunoprecipitated with antibodies against HBsAg (Hs33 mAb), VSV-G (41A1 mAb), and HCV-E1E2 (AR3A mAb) glycoproteins, as indicated, were quantified by RT-qPCR after elution. The results are expressed as percentages of input values. **f** Electron microscopy of heparin bead-purified supernatants after elution and negative staining showing large (white arrows) and small (black arrows) particles. Scale bar: 100 nm. **g** HDV RNAs, from fractions from cell supernatant samples separated on equilibrium-density gradients, were analyzed by RT-qPCR, expressed as percentages of total HDV RNA contents, or by strand-specific RT-PCR that reveals HDV genome size (below each graph). Source data are provided as a Source Data file. Error bars correspond to standard deviation. Statistical analyses (Student's t-test): $p < 0.05$ (*)

of Δp-producer cells with heparin-coated beads and we examined the eluted material by electron microscopy. We observed two types of spheres with diameters of 35–40 and 25–30 nm (Fig. 1f). The small spheres likely corresponded to subviral particles since they were also detected when VSV-G and HCV-E1E2 were expressed alone, similar to HBV GPs (Supplementary Fig. 3c, 3d), whereas the large spheres, that were only detected when HDV RNA were transcribed along with either co-expressed GP (Supplementary Fig. 3a, 3b), could correspond to VSV-Δp and HCV-Δp particles. Next, the supernatants of Δp-producer cells were subjected to equilibrium centrifugation on preformed iodixanol density gradients. Fractions collected from the gradients were assayed for density and HDV RNA by RT-qPCR (Fig. 1g). We found that HDV particles assembled with HBV GPs (noted "HDV" in the figures) exhibited a major peak of HDV RNAs at 1.12–1.14 g/mL, whereas HDV RNAs were detected at lower densities of 1.07–1.10 for VSV-Δp and of 1.08–1–13 for HCV-Δp (Fig. 1g). Finally, we found that these secreted HDV RNAs were genomic RNAs, as shown by strand-specific RT-PCR-binding assays (Fig. 1g, below density graphs). Altogether, these results indicated that heterologous envelope GPs can induce assembly of HDV particles, which are homogeneous and peak at densities likely reflecting the physicochemical features of the combination of HDV RNPs with these envelope GPs of different natures.

**HDV assembled with heterologous envelope GPs is infectious.** To determine whether the HDV particles produced with VSV-G or HCV-E1E2 envelope GPs were infectious, we performed infection assays using HDV replication-permissive Huh-106, Huh-7, and 293T cells that express different sets of virus entry receptors. At 7 days post inoculation, i.e., corresponding to the plateau of HDV RNA replication[29,32] (see below), the levels of infected cells and intracellular HDV RNAs were measured by counting HDAg-positive focus-forming units (FFU) via HDAg immunofluorescence (Fig. 2b, c) and by RT-qPCR (Fig. 2a), respectively. We found that the HDV particles produced with HBV envelope GPs could readily induce HDV RNA replication in inoculated Huh-106 cells expressing the HBV receptor, NTCP, but neither in Huh-7 nor in 293T cells, that are NTCP-negative, over the experimental thresholds provided by the "No GP" conditions, in agreement with previous studies[12,33]. Importantly, we found that the VSV-Δp and HCV-Δp particles were infectious (Fig. 2a–c). First, through RT-qPCR detection of HDV RNA in inoculated cells, we found that VSV-Δp could readily induce HDV RNA replication in the three cell types that all express the VSV-G receptor, LDLr[34,35], whereas HCV-Δp could efficiently infect Huh-106 and Huh-7 cells but less efficiently 293T cells

(Fig. 2a), in line with the differential expression of HCV receptors in either cell type[36]. Second, by using limiting dilution assays through immunofluorescence detection of HDAg (Fig. 2b), which indicated translation of HDV RNAs in inoculated cells, we confirmed that the levels of infectivity detected for the VSV-Δp and HCV-Δp particles were comparable to those of HBV GP-coated HDV particles. We deduced that all three particles type had similar specific infectivity, which is defined here by the ratio between the number of infectious viruses (measured in FFUs) and the amounts of viral RNA-containing particles (determined by RT-qPCR), with one infectious particle per 4000–7000 physical particles (Fig. 2d).

Next, to demonstrate that HDV RNA was transmitted by a bona fide HDV infectious process, we incubated producer cells with Lonafarnib, an inhibitor of prenylation that prevents HDV assembly[37,38], which requires RNP targeting to the ER membrane by farnesylation of L-HDAg[38]. We found that Lonafarnib could readily inhibit production of HBV GP-coated HDV, VSV-Δp, and HCV-Δp particles (Fig. 3a) and hence, transmission and replication of HDV RNA in inoculated cells (Fig. 3b). These results indicated that farnesyl-mediated targeting to ER or other cell membranes is required for assembly of VSV-Δp and HCV-Δp particles, suggesting that they share with HDV the same early steps, leading to production of infectious particles. Through time-course analysis, we found that cells inoculated with VSV-Δp and HCV-Δp particles accumulated over time post infection both gRNA and agRNA (Fig. 3c), which indicated that HDV RNAs could be amplified in a typical manner following entry into cells. We show that this correlated with accumulation of genomic-size HDV RNA (Fig. 3d) as well as of S-HDAg and L-HDAg proteins (Fig. 3e) at similar levels and/or ratios than for HBV GP-coated HDV particles, which indicated that full-sized HDV genomes were replicated and translated in infected cells. Altogether, these results demonstrated that HDV particles coated with the envelope GPs of VSV and HCV induce functional entry into cells and, hence, are infectious.

Then, to establish if VSV-Δp and HCV-Δp enter in the cells through the same pathways as for the parental viruses (VSV and HCV), particles were pre-incubated with antibodies that are known to neutralize VSV and HCV before their inoculation onto Huh-106 cells. The results in Fig. 4a show that the Hs33 antibody targeting the HBsAg protein readily neutralized HDV particles bearing HBV GPs but not HDV particles bearing the other GPs. Conversely, the 41A1 antibody that blocks the entry of VSV[36] neutralized VSV-Δp, whereas the AR3A antibody that neutralizes HCV[39] could only prevent infection of HCV-Δp particles. Then, we sought to block the cell receptors used by either parental virus with specific inhibitors (Fig. 4b). We found that while taurocholic

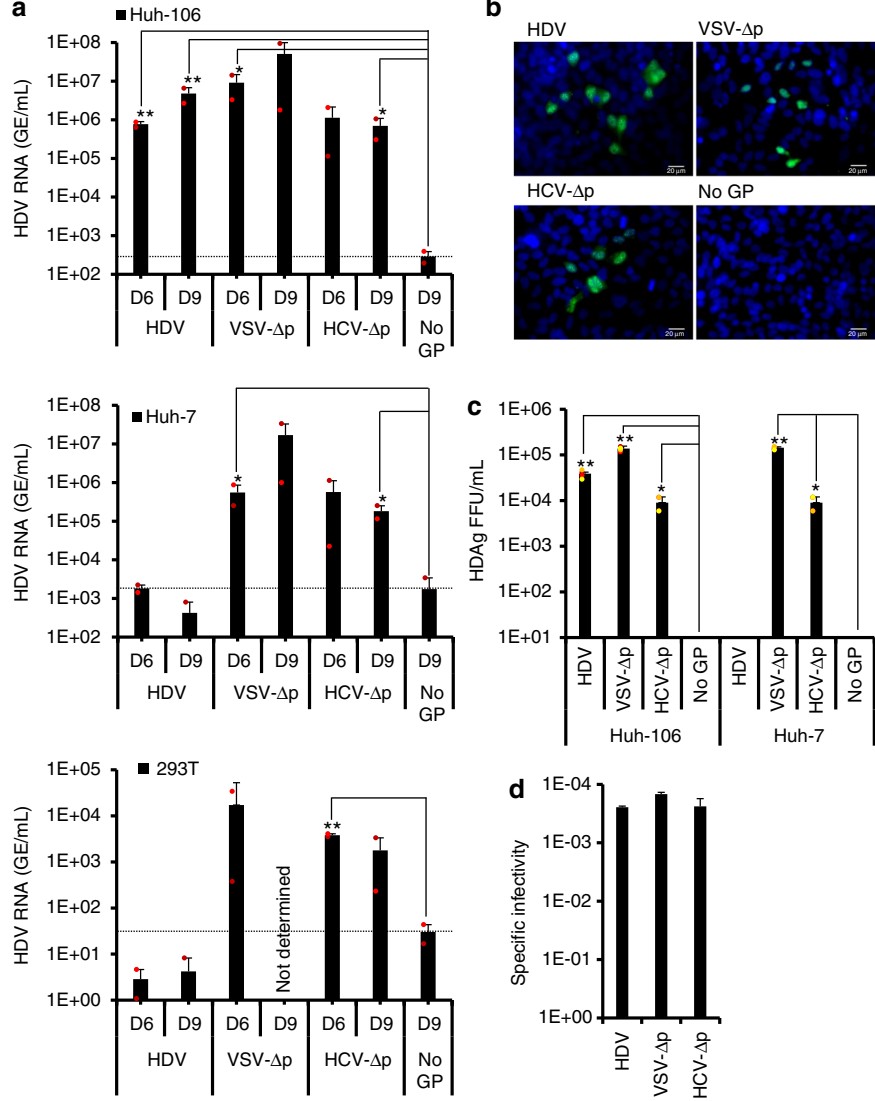

**Fig. 2** HDV particles generated with heterologous envelope glycoproteins are infectious. **a** The infectivity of virus particles produced with HBV (HDV), VSV (VSV-Δp), HCV (HCV-Δp) glycoproteins, or with no envelope glycoprotein (No GP) and harvested at day 6 or 9 post transfection (see Fig. 1a) was determined in Huh-106 (NTCP-expressing Huh-7 cells), Huh-7, or 293T cells, as indicated. Infected cells were grown for 7 days before total intracellular RNA was purified. The results of HDV RNA quantification by RT-qPCR are expressed as means ($n = 2$ independent experiments) per mL of cell lysates containing $10^5$ cells. Nd, not determined. The dotted lines represent the experimental thresholds, as defined with the "No GP" controls. **b**, **c** Huh-106 and Huh-7 cells infected by serial dilutions of supernatants containing the indicated virus particles harvested at day 9 post transfection (Fig. 1a) were fixed at 7 days post infection and stained by immunofluorescence with the SE1679 polyclonal anti-HDAg antibody before counting the foci of HDAg-positive cell colonies. The cells were counterstained with Hoechst to visualize the nuclei. Scale bars represent 20 μm (**b**). The results from colony counting are expressed as means ($n = 4$ independent experiments) of FFU per mL of cell supernatants (**c**). **d** The specific infectivity values of the indicated viruses determined in Huh-106 infected cells were calculated from the experiments shown in **c** using the infectious titers and the HDV RNA contents of the inoculums. The results show the ratios of HDAg-positive FFU induced by HDV RNA from the same inoculums. Source data are provided as a Source Data file. Error bars correspond to standard deviation. Statistical analyses (Student's $t$-test): $p < 0.05$ (*); $p < 0.01$ (**)

acid (TCA) specifically inhibited infection of Huh-106 cells with HDV particles produced with HBV GPs, as expected[12], antibodies against LDLr and CD81 (i.e., one of the HCV receptors) blocked the entry of VSV-Δp and HCV-Δp particles, respectively, though LDLr antibodies exhibited some nonspecific levels of inhibition. Altogether, the results of virus-neutralization and receptor-blocking assays indicated that the conformation of the surface of VSV-Δp and HCV-Δp particles is similar to that of parental viruses and able to mediate GP-specific cell entry through their corresponding receptors.

Thereafter, we explored if HDV particles could be produced with envelope GPs from a broader set of enveloped viruses.

Hence, we co-transfected pSVLD3 with plasmids encoding the GPs from RD114 cat endogenous virus, murine leukemia virus (MLV), human immunodeficiency virus (HIV), avian influenza virus (AIV), lymphocytic choriomeningitis virus (LCMV), human metapneumovirus (HMPV), dengue virus (DENV), and West Nile virus (WNV), which did not prevent HDV RNA replication (Fig. 5a). We detected the secretion of HDV particles induced by the GPs from HMPV, DENV, and WNV at levels similar to those of HBV GPs and at lower levels with the GPs from LCMV, though not with the GPs from the other viruses (Fig. 5b). Importantly, while no infectivity could be detected in the supernatants from the latter GPs, HDV particles enveloped with

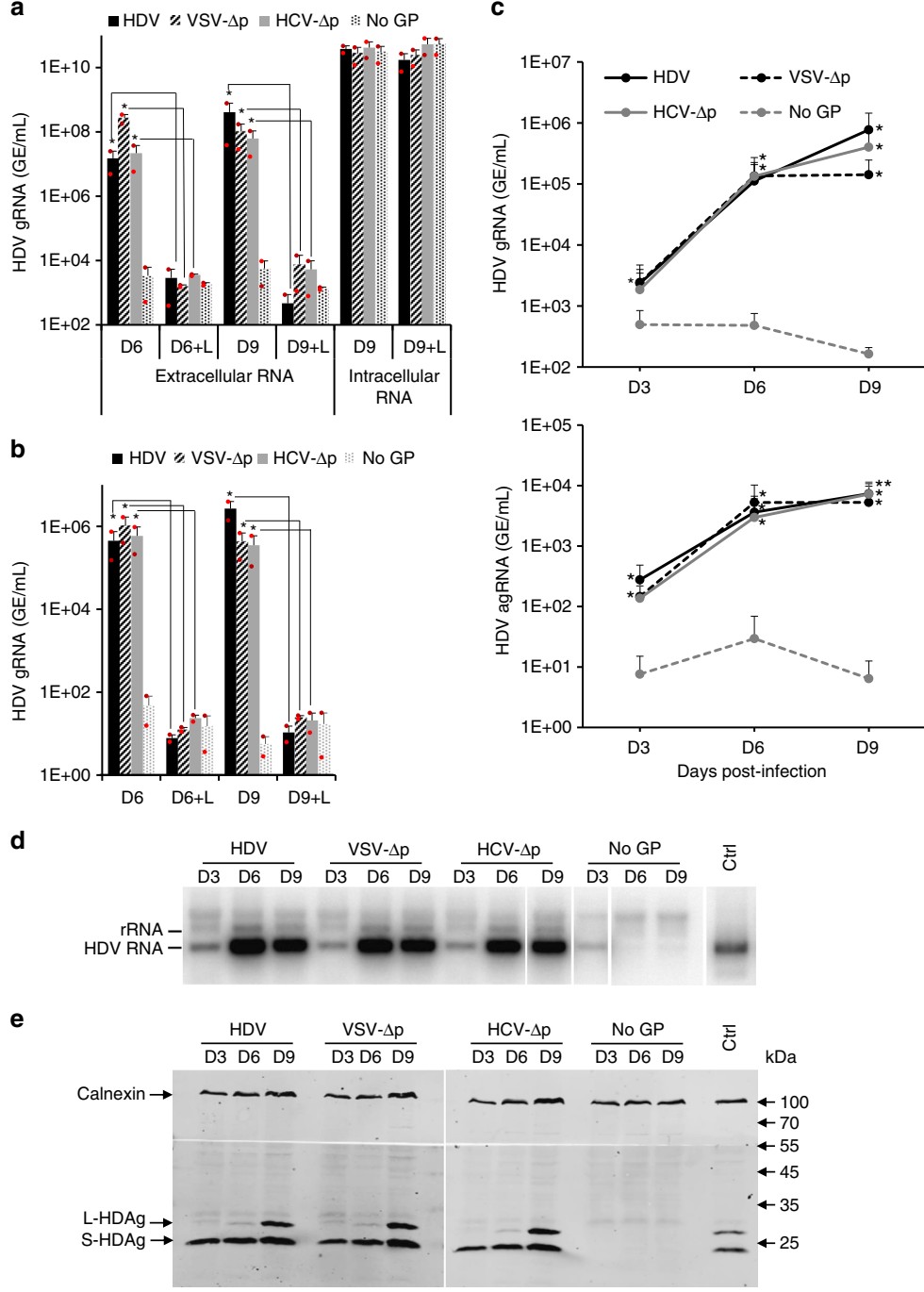

**Fig. 3** HDV, VSV-Δp, and HCV-Δp particles share an early step of assembly and induce identical HDV markers in infected cells. Huh-7 cells were co-transfected with pSVLD3 plasmid coding for HDV RNPs and plasmids coding for HBV (HDV), VSV (VSV-Δp), or HCV (HCV-Δp) envelope glycoproteins. As control, pSVLD3 was transfected without envelope proteins (No GP). **a** The transfected cells were grown in the presence (or not) of 1 mM Lonafarnib (+L), a farnesyltransferase inhibitor, until collecting at day 6 or 9 post transfection (D6 vs. D6 + L and D9 vs. D9 + L) the cell supernatants, which were filtered and inoculated to Huh-106 cells. The RNAs from producer cells and supernatants were extracted and the HDV genomes (gRNAs) were quantified by a strand-specific RT-qPCR assay. The quantification of intracellular HDV RNAs in cells producing the HDV particles at day 9 post transfection is also shown. HDV RNA levels in GE (genome equivalent) are expressed as means ($n = 2$ independent experiments) per ml of cell supernatants for extracellular RNAs or, for intracellular RNAs, per ml of cell lysates containing $10^6$ cells. **b** The inoculated cells were grown for 7 days before total intracellular RNA was purified. The results of HDV gRNA quantification by RT-qPCR are expressed as means ($n = 2$ independent experiments) per ml of cell lysates containing $10^6$ cells. **c**–**e** Huh-106 cells inoculated with the indicated viral particles were harvested at different time points post infection. The RNAs were then extracted from the lysed cells. The HDV RNAs were quantified by genomic (gRNA) (upper panel) or antigenomic (agRNA) (lower panel) strand-specific RT-qPCR assays and are expressed as means ($n = 4$ independent experiments) GE per ml of cell lysates containing $10^6$ cells (**c**). The results of a northern blot experiment using 3 μg of total cellular RNA per well that were revealed with a HDV-specific probe (**d**). Intracellular proteins were extracted and analyzed by western blot using an HDAg antibody (**e**). Control HDV RNAs ($5 \times 10^7$ GE) (**d**) or HDAg from cell lysates (**e**) were loaded on the same gels (Ctrl). Source data are provided as a Source Data file. Error bars correspond to standard deviation. Statistical analyses (Student's *t*-test): $p < 0.05$ (*); $p < 0.01$ (**)

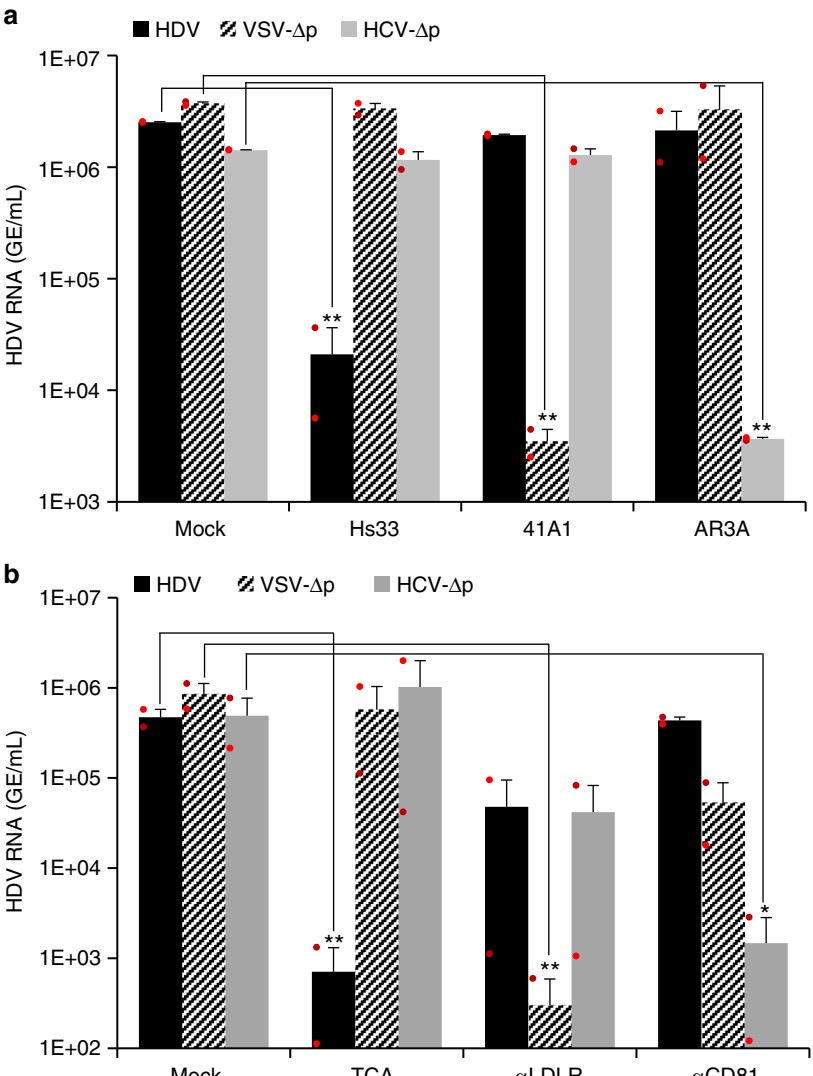

**Fig. 4** Specific glycoprotein-receptor interactions mediate cell entry of HDV particles. **a** Similar inputs of virus particles produced with HBV (HDV), VSV (VSV-Δp), or HCV (HCV-Δp) glycoproteins were incubated for 1 h at 37 °C with 100 ng/mL of neutralizing monoclonal antibodies against HBV HBsAg (Hs33 mAb), VSV-G (41A1 mAb,) and HCV-E1E2 (AR3A mAb) glycoproteins vs. no antibody (mock) before infection of Huh-106 cells. **b** Similar inputs of virus particles were used to infect Huh-106 cells that were pre-incubated for 1 h with compounds that block NTCP (TCA, taurocholic acid), LDLr (C7 mAb), and CD81 (JS-81 mAb) vs. no antibody (mock). Infected cells were grown for 7 days before total intracellular RNA was purified. The results of HDV RNA quantification by RT-qPCR are expressed as means ($n = 2$ independent experiments) per mL of cell lysates containing $10^6$ cells. Source data are provided as a Source Data file. Error bars correspond to standard deviation. Statistical analyses (Student's $t$-test): $p < 0.05$ (*); $p < 0.01$ (**)

the former GPs were infectious. They exhibited high infectivity for those enveloped with DENV GPs (Fig. 5c), similar to HBV, VSV-G, and HCV GPs (Fig. 2), but intermediate or lower infectivity for particles assembled with LCMV, HMPV, and WNV.

Finally, to extend these findings, we determined if HDV particles could be produced from non-liver cells. Hence, the pSVLD3 plasmid was co-transfected with members of the above set of GP-expression plasmids in 293T human kidney cells. Similar to production in Huh-7 cells, we found that HDV RNA could replicate in 293T cells, and that infectious HDV particles could be efficiently assembled and secreted with the HBV, VSV, HCV, DENV, WNV, and HMPV GPs (Supplementary Fig. 4), indicating that assembly and release of functional HDV with heterologous GP is not cell-type restricted.

**HDV coinfection with HCV or DENV rescues infectious HDV.** Next, to validate and extend the results of expression assays to a

more relevant infectious context, we sought to determine if HCV-Δp and DENV-Δp particles could be produced after inoculation of live HCV or DENV to cells expressing intracellular HDV RNPs. Hence, we inoculated Huh-7.5 cells producing HDV RNAs with either cell culture-grown HCV (HCVcc) or DENV at two different MOIs, which were set at suboptimal values in order to prevent virus-induced cell death. As control, we performed HBV infection assays in Huh-106 cells producing HDV[12,33].

At 5 days post inoculation with HCV, we detected intracellular HCV-NS5A and HDAg in ca. 5–10% of co-infected cells (Supplementary Fig. 5a). HCV and HDV RNAs were then quantified by RT-qPCR from cell lysates and supernatants. As shown in Fig. 6a, we could readily detect intracellular HCV RNAs in cells replicating or not HDV RNA. Identical levels of intracellular HDV RNAs of genomic size were detected in HDV-expressing cells inoculated or not with HCV (Fig. 6a). Likewise, HCV RNAs were detected in supernatants of these cells at levels that were not affected by the presence of intracellular

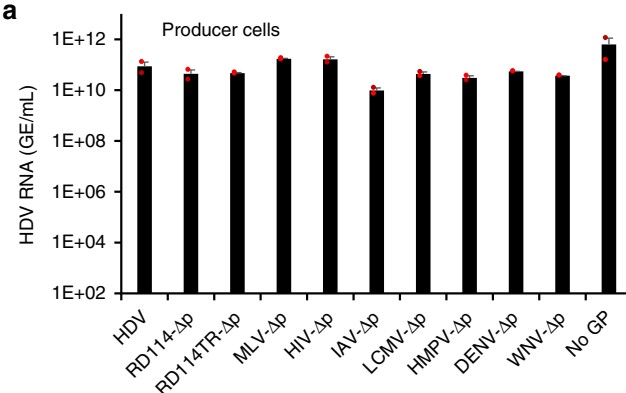

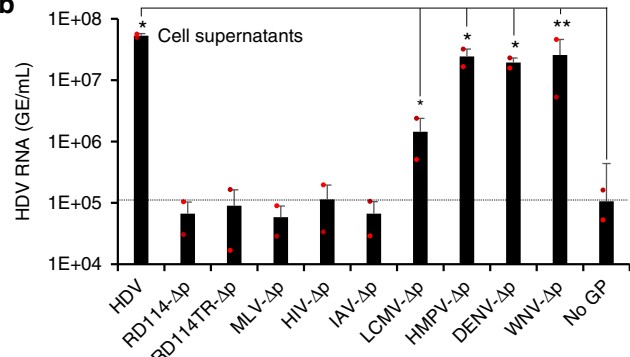

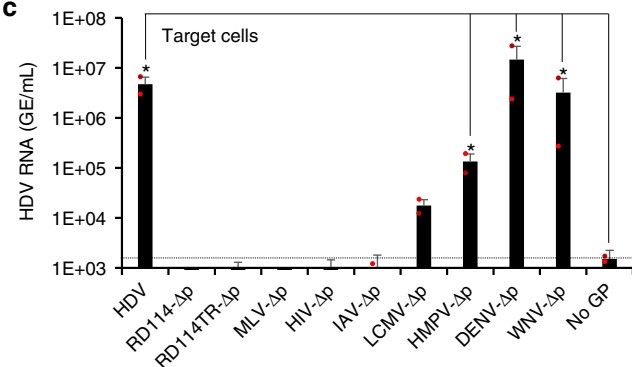

**Fig. 5** Screening of surface glycoproteins from different enveloped viruses that allow production of infectious HDV particles. Huh-7 cells were co-transfected with pSVLD3 plasmid coding for HDV RNPs and plasmids coding for HBV glycoproteins (designated "HDV") or for surface glycoproteins of the indicated enveloped viruses. The RD114TR GP is a cytoplasmic tail-modified variant of the RD114 GP that allows its trafficking to late endosomal compartments[54,55]. As control, pSVLD3 was co-transfected with an empty plasmid (referred to as "No GP"). **a** The quantification of intracellular HDV RNAs in lysates of cells at day 9 post transfection is shown. HDV RNA levels in GE (genome equivalent) are expressed as means ($n = 2$ independent experiments) per mL of cell lysates containing $10^6$ cells. **b** At day 9 post transfection, the cell supernatants were harvested, filtered, and the extracellular RNA was extracted and purified before quantifying HDV RNAs by RT-qPCR. HDV RNA levels in GE are expressed as means ($n = 2$ independent experiments) per mL of cell supernatants. **c** Huh-106 cells were incubated with the above supernatants. Infected cells were grown for 7 days before total intracellular RNA was purified. The results of HDV RNA quantification by RT-qPCR are expressed as means ($n = 2$ independent experiments) per mL of cell lysates containing $10^6$ cells. The dotted lines represent the experimental thresholds, as defined with the "No GP" controls. Note that only supernatants containing secreted HDV RNAs (**b**) allow infectivity of HDV particles containing HBV (HDV), LCMV (LCMV-Δp), HPMV (HPMV-Δp), DENV (DENV-Δp), or WNV (WNV-Δp) GPs (**c**). Source data are provided as a Source Data file. Error bars correspond to standard deviation. Statistical analyses (Student's *t*-test): $p < 0.05$ (*); $p < 0.01$ (**)

Noteworthy, the production and infectivity levels of HDV particles produced by HCV/HDV co-infected cells were similar to those of HDV particles produced with HBV as a co-infecting helper virus (Fig. 6e–h). Overall, this indicated that infectious HDV particles can be produced by coinfection with a non-HBV helper virus. To further address this, since DENV GPs could also provide helper functions for HDV RNP secretion (Fig. 5), we investigated HDV propagation from DENV/HDV co-infected Huh-7.5 cells (Fig. 6i; Supplementary Fig. 5a). We found that DENV coinfection could induce the replication and secretion of full-sized genomic HDV RNAs (Fig. 6i, j) at high levels, equivalent to those obtained via DENV GP co-expression (Fig. 5). This resulted in efficient HDV and DENV infection levels in Huh-7.5 target cells (Fig. 6k, l). Interestingly, similar results were obtained when DENV/HDV particles were inoculated in C6/36 *Aedes albopictus* mosquito cells that are permissive to DENV infection (Supplementary Fig. 6). We detected HDV (and DENV) RNAs in DENV/HDV-infected C6/36 cells (Supplementary Fig. 6d, 6e), which indicated entry and replication of HDV RNA in insect cells, though at lower levels than for Huh-7.5 cells (Supplementary Fig. 6a, 6b). Moreover, these DENV/HDV-infected C6/36 cells allowed HDV RNP assembly, secretion, and transmission to both Huh-7.5 and C6/36 naive cells (Supplementary Fig. 6f, 6g).

Overall, these results indicated that infectious HDV particles could be assembled in cells co-infected with different viruses other than HBV, and that replication and infectivity of co-infecting virus seem not affected by HDV replication.

**HCV/HDV coinfection can disseminate in vivo**. We then sought to demonstrate that HCV could propagate HDV RNPs in vivo. We generated cohorts of liver-humanized mice (HuHep-mice) derived from the FRG mouse model[40] (Fig. 7a). We retained the animals that displayed >15 mg/mL of human serum albumin (HSA), which corresponded to 40–70% of human hepatocytes in the liver[41]. In agreement with previous reports[41,42], these animals supported HBV (Group#1) and HCV (Group#5) infection for several months (Fig. 7b; see

HDV RNA (Fig. 6b). Notably, we found that extracellular HDV RNAs could be readily detected only in the supernatants from cells co-infected with HDV and HCV (Fig. 6b), indicating that HCV infection can provide helper functions for assembly and secretion of HDV particles. These particles incorporated full-sized genomic HDV RNA, as shown by strand-specific RT-PCR assays (Fig. 6b, below the graphs). Next, we determined the infectivity of virus particles by measuring intracellular HCV and HDV RNAs in Huh-7.5 target cells 7 days after their inoculation with the producer cell supernatants. We detected HCV RNAs in these target cells, reflecting the presence of infectious HCV particles in the supernatants of HCV-(co)infected cells, at similar levels whether or not HDV genome was co-expressed in the producer cells. Importantly, we found that the HDV particles produced from HCV/HDV co-infected cells were infectious (Fig. 6c), as we could readily detect HDV RNAs in these target cells well over the experimental threshold provided by the control conditions. Corroborating these results, cells that were co-infected by both HCV and HDV or that were mono-infected by either virus were observed by immunofluorescence (Fig. 6d).

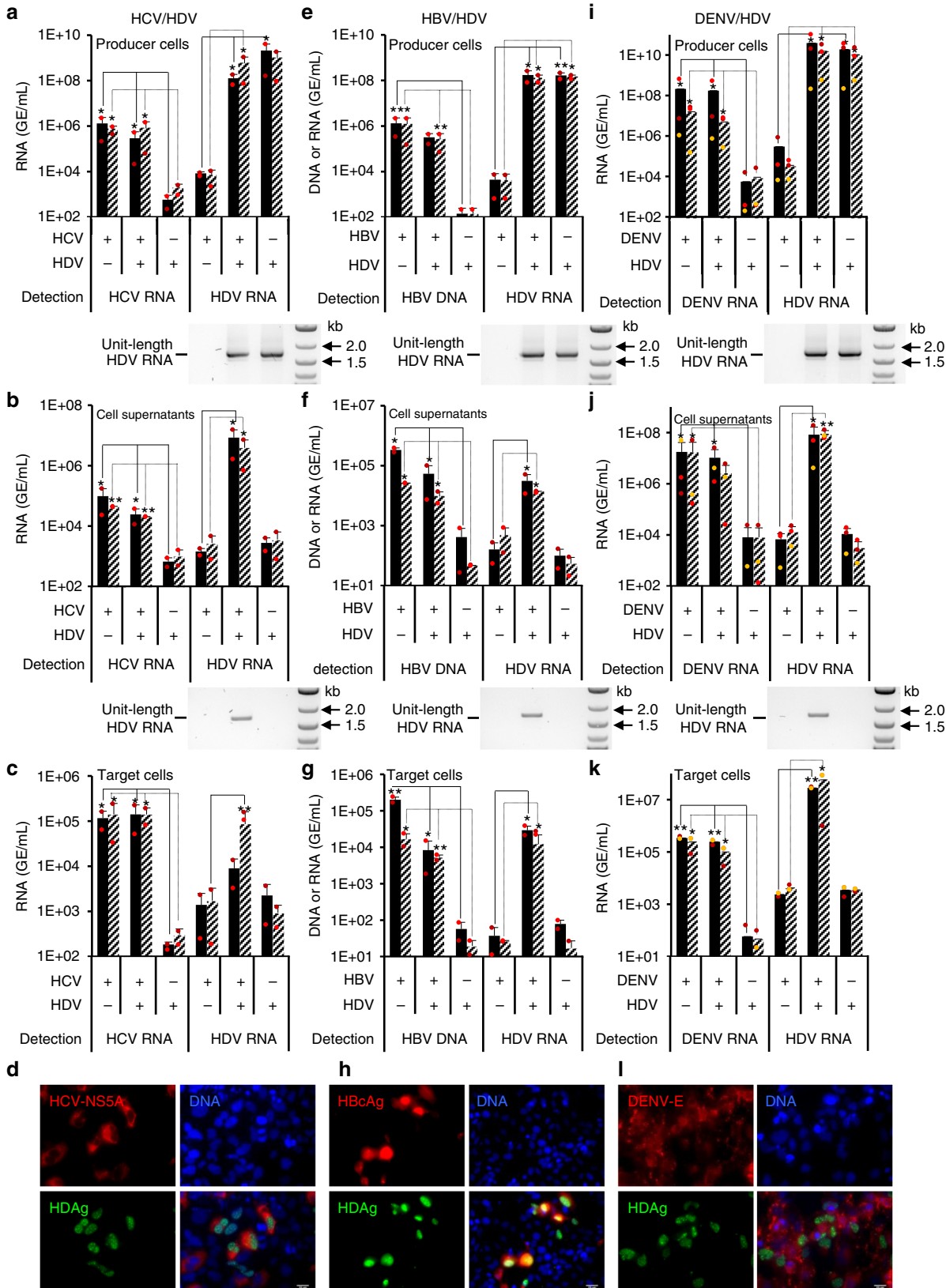

Supplementary Fig. 7a for individual mice). In contrast, inoculation of HuHep-mice with "helper-free" HDV, i.e., HDV particles produced with HBV GP-expression plasmid (Fig. 1), did not lead to HDV viremia, as shown by RT-qPCR values in infected animal sera that were identical to those detected in the non-infected HuHep-mice control group (Group#9: HDV vs.

Group#10: Mocks; Supplementary Fig. 7a). The other groups of HuHep-mice (5–8 animals each) were inoculated with either "helper-free" HDV followed by HCV 4 weeks later (Group#7), HCV followed by "helper-free" HDV (Group#6), or both HCV and "helper-free" HDV simultaneously (Group#8). HDV RNAs were detected in animals of the three latter groups within a few

**Fig. 6** HDV RNA-producing cells infected with HCV and DENV secrete infectious HDV particles. Huh-7.5 (**a**, **i**) or Huh-106 (**e**) cells producing HDV RNAs were inoculated with high (black bars) vs. low (hatched bars) MOIs of live HCV (MOI = 0.01 and 0.1 FFU/cell; **a**), HBV (MOI = 20 and 200 GE/cell; **e**) or DENV (MOI = 0.01 and 0.1 FFU/cell; **i**) particles. Supernatants and lysates from these cells were harvested at day 5 (HCV, DENV) and day 7 (HBV) post infection. HDV-expressing cells without subsequent infection (referred to as "HDV") as well as naive cells only infected with HCV, HBV, or DENV, as indicated in legends below each graph, were used as controls. Supernatants from HCV/HDV (**b**), HBV/HDV (**f**), or DENV/HDV (**j**) co-infected cells or corresponding control cells were used to infect Huh-7.5 (**c**, **k**) or Huh-106 (**g**) cells. Infection levels were assessed at day 7 post infection. Nucleic acids present in filtered cell supernatants (**b**, **f**, and **j**) and lysates of producer (**a**, **d**, and **g**) or target cells (**a**, **f**, and **i**) were extracted and purified for quantification of HDV (**a–c**, **e–g**, and **i–k**) and HCV RNA (**a–c**), HBV DNA (**e–g**), or DENV (**i–k**) RNA by qPCR. The results expressed in GE (genome equivalent) are displayed as means ($n = 2$ (**a–c**, **e–g**) or $n = 3$ (**i–k**) independent experiments) per mL of cell supernatants for extracellular nucleic acids or, for intracellular nucleic acids, per mL of cell lysates containing $10^6$ cells. Extracted RNAs were reverse-transcribed and were PCR-amplified with HDV-specific primers to reveal the size of transcribed HDV genomes (HDV RNA unit length), as shown below the graphs. Huh-7.5 (**d**, **l**) or Huh-106 (**h**) cells co-infected with HDV and HCV (**d**), HBV (**h**), or DENV (**l**) were fixed 7 days after infection, stained for HDAg and HCV-NS5A, HDAg and HBcAg, and HDAg and DENV-E, respectively, and counterstained with Hoechst to visualize the nuclei. HDAg (green channel), HCV-NS5A, HBcAg, DENV-E (red channels), and nuclei (blue channel) were then visualized by immunofluorescence. Scale bars represent 20 µm. Source data are provided as a Source Data file. Error bars correspond to standard deviation. Statistical analyses (Student's t-test): $p < 0.05$ (*); $p < 0.01$ (**)

weeks after inoculation. All HCV-positive animals of these groups were also positive for HDV (Fig. 7b; Supplementary Fig. 7a) and secreted HDV RNA of genomic size was detected in the sera (see examples for two animals/group in Supplementary Fig. 7b). We obtained qualitatively comparable results in HuHep-mice co-infected with HDV and HBV (Fig. 7a, b, Group#2, #3, and #4; Supplementary Fig. 7a, 7b). Of note, similar results were obtained in another cohort of HuHep-mice in which HDV was inoculated 1 week after HCV (Supplementary Fig. 8). Altogether, these results indicated that HDV can be propagated in vivo by different virus types, including HCV.

## Discussion

Satellite viruses are scarcely found in animal viruses in contrast to their profusion in plant viruses. Only two representative satellite viruses are known currently in human viruses and include HDV and adeno-associated virus (AAV), which uses helper functions of e.g., adenovirus or herpes simplex virus at the level of replication of its genome, unlike for HDV. Indeed, HDV has been described as a satellite virus of HBV, a liver-specific human pathogen that provides its surface GPs to induce envelopment and secretion of HDV RNPs, as well as transmission to other cells via HBV cell entry factors. In vivo, HDV has been found to be associated with HBV in >5–10% of the ca. 250 million HBV-infected individuals[43] worldwide.

A specific feature of HBV is the assembly and secretion of different types of viral particles. While the three HBV envelope GPs, S-HBsAg, M-HBsAg, and L-HBsAg, induce the secretion of bona fide virions incorporating HBV capsid and DNA genome through the ESCRT assembly and budding machinery in cellular multivesicular bodies (MVB)[44,45], they also induce a particularly abundant formation of HBV nucleocapsid-free subviral particles (SVPs) at a pre-Golgi membrane that are subsequently exported through the cell secretory pathway[46]. The latter type of particles is exploited by HDV through a process allowing binding of its RNP to a cytosolic determinant of the HBV envelope GPs[29,38,47–49]. Hence, as HBV SVPs outnumber by ca. 4 orders of magnitude the HBV virions[50], HDV RNPs are particularly efficiently coated and secreted with the HBV envelope GPs, with titers that can reach up to $10^{11}$ HDV virions per mL of serum, and are consequently transmitted to the liver, which explains why HDV and HBV share tropism to human hepatocytes.

Yet, genetically, HDV belongs to a group of infectious agents that are related to plant viroids and that are completely distinct from HBV. As HDV efficiently replicates in different tissues and species[23], here we raised the hypothesis that it may have arisen from and/or conceivably still infects hosts independently of HBV. To formally address this possibility, we questioned whether different

enveloped viruses, totally unrelated to HBV and HDV themselves, could provide both assembly and entry functions to HDV particles. By testing GPs from ten different virus genera, we demonstrate that HDV RNPs could be enveloped by GPs from six of these non-HBV particles and could produce infectious HDV particles.

The nature of the determinant(s) and mechanism(s) allowing HDV assembly with these unconventional GPs remains to be unraveled. Noteworthy, a farnesylation signal located at the C-terminus of L-HDAg anchors the HDV RNP to the ER membrane[38], the site where, by definition, envelope GPs are generally synthesized and translocated. Such early assembly events of HDV production seem also to be used for assembly of HDV RNP with alternative GPs, such as VSV-G and HCV-E1E2 GPs, since inhibition of this pathway by Lonafarnib, a farnesyltransferase inhibitor that is currently in phase-IIa clinical trial[51], could readily prevent production and transmission of HDV, VSV-Δp, and HCV-Δp particles (Fig. 3). As for conventional HDV particles assembly, i.e., associated with HBV GPs, S-HBsAg is necessary and sufficient for assembly of HDV, although incorporation of L-HBsAg is required for infectivity[52]. Previous studies have described a crucial determinant of HDV envelopment, consisting of a conserved tryptophan-rich motif present in the cytosolic side of the S-HBsAg that acts as an HDV matrix domain and binds a poorly conserved proline-rich C-terminal sequence located before the farnesylation site of L-HDAg[48]. Yet, such a tryptophan-rich motif is likely not present in the heterologous GPs that induce efficient HDV release, such as e.g., VSV-G, HCV-E1E2, and DENV-PrME, inferring that a specific interaction between HDV RNP and these envelope GPs is highly improbable. Rather, this indicates that besides such specific HDV/HBV interaction allowing HDV transmission and subsequent pathogenesis, other determinant(s) of envelopment of HDV RNPs must exist.

How viruses in general exploit or subvert cellular envelopment processes and machineries is of major interest. Budding mechanisms vary widely for different virus families and there are few common principles that govern these events. Particularly, the assembly and budding of enveloped virus particles is a complex and multistep process that involves the simultaneous recruitment of viral proteins, surface GPs and inner structural proteins, and nucleic acids to varying assembly sites. Such sites can be localized either at the plasma membrane (e.g., HIV) or in the lumen of diverse intracellular membranes (e.g., HCV, DENV), such as the ER as well as the nuclear envelope, the intermediate or pre-Golgi compartment, the Golgi cisternae and trans-Golgi network, and the endosomes. Alternatively, assembly sites can be generated via specific virus-induced membranous structures or compartments[53]. Accordingly, a critical determinant of GP incorporation in the envelopes of retrovirus particles allows intracellular trafficking of GP and colocalization with nucleocapsids, although

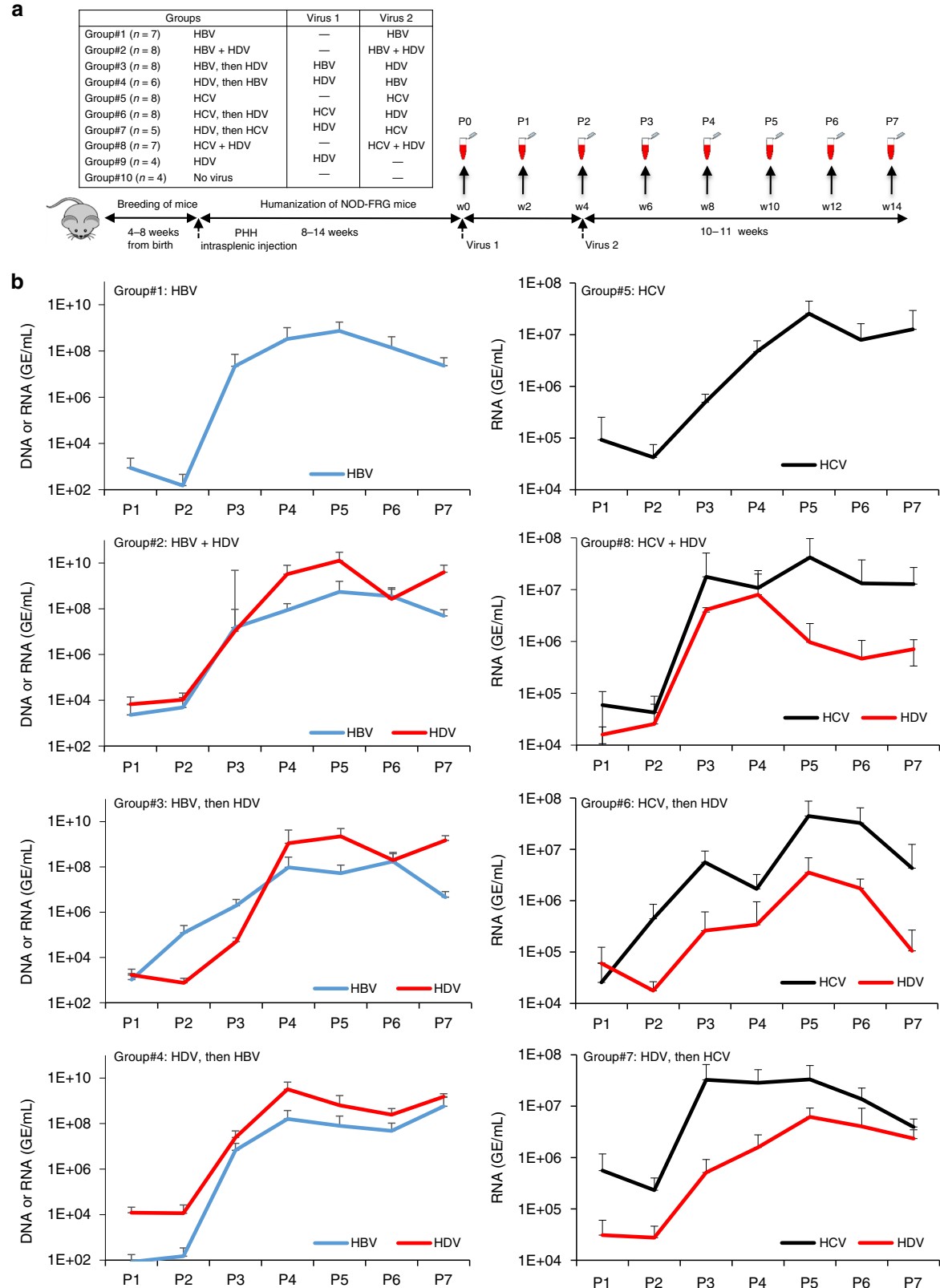

**Fig. 7** HCV propagates HDV particles in vivo. Four- to eight-week-old NOD-FRG mice were engrafted with primary human hepatocytes (PHH). After ca. 2–3 months, the animals displaying HSA levels >15 mg/mL were split into 10 different groups ($n = 4$ to $n = 8$ independent animals) that were infected with HDV ($10^7$ GE/mouse) and/or HCV ($1.5 \times 10^5$ FFU/mouse) or HBV ($10^8$ GE/mouse), as shown in the schedule (**a**). At different time points post infection, blood samples (50 µl) were collected and the viremia in sera was monitored by qPCR on the genomes of the indicated viruses (GE/mL of serum) (**b**). The graphs show the mean results of viremia of HDV (red lines), HBV (blue lines), and HCV (black lines). See results of individual mice as well as of control groups, inoculated with HDV only (Group#9: HDV) or with PBS (Group#10: Mocks) in Supplementary Fig. 7. Source data are provided as a Source Data file. Error bars correspond to standard deviation

signals modulating direct as well as indirect interactions of the GP cytoplasmic tails with nucleocapsid components have been described[54,55]. Thus, different scenarios could explain how HDV RNPs may incorporate non-HBV glycoproteins. Besides factors allowing colocalization and/or interactions between GP and nucleocapsids, the budding and subsequent envelopment of viral particles requires the curvature and scission of the host membrane concomitant with the inclusion of nucleocapsid components. The driving force for budding can be provided by the nucleocapsid itself, via specific inner structural proteins (e.g., Gag precursor of HIV) that "pushes" a virion membranous bud through the cytoplasmic side of a membrane. Alternatively, budding can be driven by an envelope GP that, by forming a symmetric lattice (e.g., prME GP of flaviviruses) or alternatively, a cellular vesiculation (e.g., G protein of VSV), "pulls" the membrane, creates a bud in which the nucleocapsid can be incorporated. Although there are many subtle variations and/or combinations between these two main models[56], it is intriguing that the enveloped viruses that induce an efficient release of HDV particles (this report) are known to form subviral particles, i.e., nucleocapsid-free vesicles coated with envelope GPs, which typically pertain to the "pull" model of virion assembly/budding. Indeed, in addition to their own infectious particles, HBV[46], VSV[57], HCV[58,59], DENV[60], and WNV[61] can release their GPs in sediment, vesicular forms, which, at least for HBV, HCV and flaviviruses are assembled and released in the ER lumen or ER-derived compartment. Conversely, the GPs from retroviruses such as RD114, MLV or HIV, and influenza virus are released from the plasma membrane and/or late endosomes upon incorporation at the surface of infectious virions[54] but they have not been described to form SVPs, a characteristic compatible with their inability to assemble HDV particles (Fig. 5). It is puzzling that VSV-G, which induces the formation of infectious virions from the plasma membrane[62], allows efficient HDV particles release. Yet, we cannot exclude that formation of VSV-G subviral particles may also occur in the lumen of the ER or, alternatively, that HDV RNP could be targeted beneath the plasma membrane, in addition to that of the ER.

Importantly, we show that the formation of infectious HDV particles with unconventional GPs could also occur via coinfection with live viruses different from HBV, as shown by their release from HCV/HDV or DENV/HDV co-infected cells. We presume that this occurs through the same mechanisms of HDV RNP assembly and envelopment in SVPs formed by either virus type, as proposed above. Noteworthy, our results reveal that HDV particles can be propagated by HCV in experimentally co-infected mice, indicating that, in this in vivo setting at least, HDV can be a satellite of a virus genus totally unrelated to HBV. This raises the possibility that in nature, HDV could be associated with different virus types, including human viral pathogens, which could possibly favor previously unappreciated HDV transmission scenarios and modulate their pathogenicity. Indeed, hepatitis D is a most aggressive form of hepatitis that affects ca. 15–20 million persons worldwide, with high disparities around the world[43]. While the clinical course of acute HDV infection is hardly distinguishable from acute hepatitis B[63], chronic HDV infection worsens liver diseases caused by HBV, even though HBV replication is often suppressed[64–66]. Longitudinal cohort studies have shown that chronic hepatitis D induces a threefold higher risk of progression to cirrhosis as compared with patients infected by HBV only[67]. HDV replicates in hepatocytes and the pathologic changes it induces are limited to the liver, which are characterized by hepatocyte necrosis and inflammatory infiltrates that may correlate with intrahepatic replication levels[68]. Thus, since HDV RNA can persist in the liver in the absence of HBV for at least 6 weeks[69], its propagation could be triggered upon superinfection

by other hepatitis viruses such as HCV as well as other viral infections. Why HDV infection in patients has not been reported in association with HCV infection is uncertain and raises interesting scenarios. It is possible that direct or indirect (e.g., immune-mediated) interference mechanisms may impede in the long-term HDV/HCV co-infections in vivo, though this may occur in different contexts. For example, HDV-induced activation of the innate immune response, which is known to have little/no effect on HDV itself[70,71], may impact several markers of coinfection, such as for HCV that is interferon-sensitive in contrast to HBV[72,73]. Thus, what might determine eventual successful vs. sporadic transmission and propagation of HDV with non-HBV helper viruses would reside in the balance between biochemical and virological compatibility of HDV RNP with the GPs of these helper viruses vs. potential immunological mechanisms of interference, though the immune status of individuals, such as immune suppression, may also favor transmission of such HDV co-infections. Overall, our demonstration that unconventional cell transmission of HDV is experimentally possible in vivo warrants that studies be conducted in infected individuals.

## Methods

**Cells**. Huh-7[49] hepatocarcinoma and Huh-106[33] (a subclone of NTCP-expressing Huh-7 cells) cells were grown in William's E medium (Invitrogen, France) supplemented with nonessential amino acids, 2 mM L-glutamine, 10 mM HEPES buffer, 100 U/mL of penicillin, 100 μg/mL of streptomycin, and 10% fetal bovine serum (FBS). Huh-7.5 cells (kind gift of C Rice) and 293T kidney (ATCC CRL-1573) cells were grown in Dulbecco's modified minimal essential medium (DMEM, Invitrogen) supplemented with 100 U/mL of penicillin, 100 μg/mL of streptomycin, and 10% FBS. The C6/36 *Aedes albopictus* cells (ATCC CRL-1660) were grown in DMEM medium supplemented with 100 U/mL of penicillin, 100 μg/mL of streptomycin, L-glutamine, and 10% FBS at 28 ºC.

**Plasmids**. pSVLD3 plasmid encodes HDV RNP[27,29]. Plasmids pT7HB2.7 for HBV[29], phCMV-VSV-G for vesicular stomatitis virus (VSV), phCMV-JFH1-E1E2 for hepatitis C virus (HCV), phCMV-RD114 and phCMV-RD114TR for cat endogenous virus, phCMV-MLV-A for amphotropic murine leukemia virus (MLV), phCMV-HIV for human immunodeficiency virus (HIV), phCMV-NA and phCMV-HA for avian influenza virus (AIV), phCMV-LCMV for lymphocytic choriomeningitis virus (LCMV), phCMV-FgsHMPV for human metapneumovirus (HMPV), phCMV-PrME for dengue virus (DENV), and West Nile virus (WNV) encode the envelope surface glycoproteins of the indicated viruses[36,74,75].

**Antibodies**. The HDAg antigen was detected with the SE1679 rabbit polyclonal antibody for western-blot and immunofluorescence experiments. The human anti-E2 AR3A[39] (kind gift from M Law), mouse anti-VSV-G 41A1[58], and mouse anti-HBsAg Hs33 (Cat # GTX41723, GeneTex) monoclonal antibodies (mAb) were used in neutralization and immunoprecipitation assays. The mouse anti-CD81 JS-81 (Cat # 555675 BD Pharmingen) and anti-LDLr C7 (Cat # sc-18823, Santa Cruz Biotechnology) mAbs were used for receptor-blocking experiments. The mouse anti-DENV-E 3H5 mAb (kind gift from P Desprès), the mouse anti-NS5A 9E10 mAb (kind gift of C Rice), and the human anti-HBcAg (from an anti-HBcAg-positive and anti-HBsAg-negative patient) serum were used for immunofluorescence.

**HDV particle production and infection**. Huh-7 cells were seeded in 10-cm plates at a density of 10^6 cells per plate and were transfected with a mixture of 2.5 μg of pSVLD3 plasmid and 10 μg of plasmid, allowing the expression of surface envelope glycoproteins of the above-mentioned viruses[29] using FuGENE 6 transfection reagent (Promega). Transfected cells were grown for up to 9 days in primary hepatocyte maintenance medium containing 2% FBS and 2% DMSO to slow cell growth[76]. Supernatants of virus-expressing cells were separated from the producer cells, filtered through 0.45-μm-pore filters, and were analyzed by RT-qPCR for detection of HDV RNA[28], using the methodologies and primers described below. These supernatants were also used for infection experiments in Huh-106 and other target cells, which were seeded in 48-well plates at a density of $1.5 \times 10^4$ cells per well.

Transfected or infected cells were cultured in primary hepatocyte maintenance medium containing 2% FBS and 2% DMSO following infection to slow cell growth. Infectivity of viral particles was assessed 7 days post infection by RT-qPCR of HDV RNA isolated from cell lysates or by determining focus-forming units in ethanol-fixed plates using HDAg antibodies. For neutralization and receptor-blocking experiments, 100 ng/mL of antibodies were incubated with virus particles for 1 h at 37 ºC before addition to the cells.

For purification of viral particles, 10 mL of producer cell supernatants were harvested, filtered through a 0.45-µm filter, and centrifuged at 32,000 rpm for 4 h at 4 °C on a 30% sucrose cushion with a SW41 rotor and Optima L-90 centrifuge (Beckman). Pellets were resuspended in 100 µL of TNE (50 mM Tris-HCl, pH 7.4, 100 mM NaCl, and 0.1 mM EDTA) prior to use for immunoprecipitation and western blot of HDAg or for northern blot of RNAs.

For inhibition of farnesyltransferase in producer cells, we used Lonafarnib (Sigma-Aldrich), an inhibitor of prenylation that prevents HDV assembly[37,38]. Following transfection with pSVLD3 plasmid and GP expression plasmid, as described above, Huh-7 cells were maintained in a daily-changed medium supplemented with 0.2% DMSO and 400 mM DTT alone or in the presence of 1 mM Lonafarnib. The cell supernatants were used for infection experiments in Huh-106 as described above.

**RT-qPCR detection of HDV RNAs**. Total RNA from serum, filtrated cell supernatant, or from virus producer or infected cells washed with phosphate-buffer saline (PBS) was extracted with TRI Reagent according to the manufacturer's instructions (Molecular Research Center) and treated with RNAse-free DNase (Life Technologies). RNAs were reverse-transcribed using random oligonucleotide primers with iScript cDNA synthesis kit (Bio-Rad) before quantification by qPCR, as described below.

For strand-specific HDV RNA RT-qPCR[30,31], extracted RNAs were reverse-transcribed with High-Capacity cDNA Reverse Transcription kit (Applied Biosystems) to amplify either genomic or antigenomic cDNAs by using primers DSg: 5′-CCGGCTACTCTTCTTTCCCTTCTCTCGTC for genomic-sense cDNA synthesis and DSag: 5′-CACCGAAGAAGGAAGGCCCTGGAGAACAA for antigenomic-sense cDNA synthesis. The qPCR assay was then performed, as described below.

The genomic and antigenomic HDV RNAs used as standards for this strand-specific RT-qPCR assay were obtained by in vitro transcription of HDV DNA amplicons flanked by T7 promoters. The full-length HDV amplicons were amplified by PCR from pSVLD3 plasmid with primers T7HD 687–706: 5′-*CAATTCTAATACGACTCACTATAGGGAGAA* GGCCGGCATGGTCCCAGCC TC (with *T7 promoter* sequences) and HDVgR: 5′-ATCAGGTAAGAAAGGA TGGAACGCGGACCC for the amplicon allowing synthesis of the genomic HDV RNA standards and, for the amplicon allowing synthesis of the antigenomic HDV RNA standards, with primers HDVgF: 5′-GGCCGGCATGGTCCCAGCCTC and AgT7HD 685–656: 5′-*CAATTCTAATACGACTCACTATAGGGAGAA*A TCAGGT AAGAAAGGATGGAACGCGGACCC (with *T7 promote*r sequence). Either amplicon was transcribed from T7 promoters using a commercially available kit according to the manufacturer's instructions (RiboMAX™ Express T7, Promega). The full-length linear RNAs were treated with RNase-free DNase, followed by RNA purification (GeneJET RNA purification kit, Thermo Fisher Scientific) and quantified using a Nanodrop device (Thermo Fisher Scientific). These artificial genomic or antigenomic HDV RNAs were diluted in RNase-free water and stored at −80 °C in single-use aliquots and were used as calibration standards for strand-specific HDV RNA RT-qPCR[31,32,77]. The copy numbers in genomic or antigenomic HDV RNAs extracted from cells or supernatants were quantified using 10-fold dilution series of either genomic or antigenomic HDV RNA standards processed in parallel. We deduced that $10^6$ HDV RNA molecules (genomic or antigenomic) are equal to 1 pg of the corresponding HDV RNA standard. The specificity of the strand-specific RT-qPCR assay was investigated by the quantification of genomic and antigenomic RNA standards (artificial RNA) with the correct primer and the respective opposite primer (genomic primer on antigenomic HDV RNA standard and vice versa) (Supplementary Fig. 1f). Unspecific reverse transcription with the opposite primer occurred, though in a very limited extent (<0.000001%).

The following specific oligonucleotides were then used for HDV cDNA quantification:[31,32] forward (Kuo F: 5′-GGACCCCTTCAGCGAACA) and reverse (Kuo R: 5′-CCTAGCATCTCCTCCTATCGCTAT) primers. The qPCR was performed using FastStart Universal SYBR Green Master (Roche Applied Science) on a StepOne Real-Time PCR System (Applied Biosystems).

As an internal control of extraction, in vitro-transcribed exogenous RNAs from the linearized Triplescript plasmid pTRI-Xef (Invitrogen) were added into the samples prior to RNA extraction and quantified with specific primers (Xef-1a 970L20: 5′-CGACGTTGTCACCGGGCACG and Xef-1a 864U24: 5′-ACCAGGCATGGTGGTTACCTTTGC). All values of intracellular HDV RNAs were normalized to *GAPDH* gene transcription. For GAPDH mRNA quantification, we used as forward primer, hGAPDH 83U: 5′-AGGTGAA GGTCGGAGTCAACG and as a reverse primer, hGAPDH 287 L: 5′-TGGAAG ATGGTGATGGGATTTC.

**RT-PCR assays**. Total RNA from cells or supernatants was extracted with TRI Reagent according to the manufacturer's instructions (Molecular Research Center). RNAs were reverse-transcribed with SuperScript III reverse transcriptase kit (Invitrogen) using a strand-specific primer HDV gRNA: 5′-ATCAGGTAAGAAAGGA TGGAACGCGGACCC that detects the genomic HDV RNA and allows amplification of antigenomic-sense cDNA. The reverse-transcribed cDNA products were used to perform a PCR using the following specific oligonucleotides to amplify the unit-length HDV genome: forward (HDVgF: 5′-GGCCGGCATGGTCCCAGCCTC) and

reverse (HDVgR: 5′-ATCAGGTAAGAAAGGATGGAACGCGGACCC) primers. The PCR bands were visualized on propidium iodide-stained agarose gels.

**Northern blots**. The purified RNA was subjected to electrophoresis through a 2.2 M formaldehyde, 1.2% agarose gel, and transferred to a nylon membrane. The membrane-bound RNA was hybridized to a $^{32}$P-labeled RNA probe specific for genomic HDV RNA[49]. Quantification of radioactive signals was achieved using a phosphorimager (BAS-1800 II; Fuji). Uncropped and unprocessed scans of all blots are provided in the Source Data file.

**Western blots**. The proteins from pelleted cell supernatants or extracted from total cell lysates were denatured in Laemmli buffer at 95 ºC for 5 min and were separated by sodium dodecyl sulfate polyacrylamide gel electrophoresis and then transferred to nitrocellulose membranes (GE Healthcare). Membranes were blocked with 5% nonfat dried milk in PBS and incubated at 4 °C with the SE1679 rabbit anti-HDAg serum at a 1/500 dilution in PBS–0.01% milk, followed by incubation with a IRdye secondary antibody (Li-Cor Biosciences). Membrane visualization was performed using an Odyssey infrared imaging system CLx (LI-COR Biosciences). Uncropped and unprocessed scans of all blots are provided in the Source Data file.

**Immunoprecipitation of HDV particles**. For immunoprecipitation, 50 µL of Dynabeads Protein G (Thermo Scientific) bound with 10 µg of anti-HBsAg Hs33, anti-E2 AR3A, or anti-VSV-G 41A1 mAbs were incubated for 1 h at room temperature with purified virus particles. The beads were then washed three times with 1 mL of PBS with 0.02% Tween-20. The RNA was extracted from the complex with Tri Reagent and detected by RT-qPCR.

**Equilibrium-density gradients**. One milliliter of cell supernatant containing virus particles harvested at 9 days post transfection was loaded on top of a 3–40% continuous iodixanol gradient[58] (Optiprep, Axis Shield). Gradients were centrifuged for 16 h at 4 °C in Optima L-90 centrifuge (Beckman). Thirteen fractions of 900 µl were collected from the top and used for refractive index measurement and RNA quantification, as described above.

**Electron microscopy**. One milliliter of cell supernatant containing virus particles harvested at 9 days post transfection was mixed with 100 µL of heparin–agarose beads (Sigma) preequilibrated with 10 mM Tris-HCL and 100 mM NaCl buffer (pH 8). Unbound particles were washed off five times with 10 mM Tris-HCL, 200 mM NaCl buffer (pH 8), and the particles were eluted from the heparin–agarose beads with 10 mM Tris-HCL, 800 mM NaCl buffer (pH 8). For negative staining in electron microscopy, 5 µL of sample solution was applied onto a glow-discharged EM grid coated with amorphous carbon. After 1 min of sample adsorption, the excess solution was blotted away using a piece of filter paper and the grid was put onto a drop of 1% (w/v) sodium silicotungstate staining solution. After 30 s, excess stain solution was blotted away as before and the grid was dried in air. The samples were examined using a transmission electron microscope Philips CM120 operating at 120 kV.

**Coinfection assays**. Huh-7.5 cells seeded in six-well plates at a density of $8 × 10^4$ cells per well producing HDV RNAs were superinfected 3 days later cells with Jc1 HCVcc, HBV, or DENV live, helper virus particles[78,79]. Lysates and supernatants of infected cells were harvested at 5 days post infection from the producer cells and were analyzed by qPCR for detection of HDV[28], HCV[58], HBV[78], and DENV nucleic acids. The supernatants containing HDV and either helper virus particles were used for infection experiments in relevant target cells. Infectivity was assessed at 7 days later by qPCR of HDV (see above) and of helper virus RNAs or DNAs isolated from cell lysates, using the following specific oligonucleotides: for HCV, forward HCV U147: 5′-TCTGCGGAACCGGTGAGTA and reverse HCV L277: 3′-TCAGGCAGTACCACAAGGC primers; for HBV, forward HBV-SUF: 5′-TCCCAGAGTGAGAGGCCTGTA and reverse HBV-SUR: 5′-ATCCTCGAGAA GATTGACGATAAGG primers; and for DENV, forward DENV NSF: 5′-ACCT GGGAAGAGTGATGGTTATGG and reverse DENV NSR: 5′-ATGGTCTCTGG TATGGTGCTCTGG primers.

**Immunofluorescence**. Producer or infected cells were fixed with 4% paraformaldehyde (Sigma-Aldrich, France) for 15 min and permeabilized with 0.1% Triton X-100 (Sigma-Aldrich) for 7 min. Fixed cells were then saturated with 3% bovine serum albumin (BSA)/PBS for 20 min and incubated for 1 h with primary antibodies diluted in 1% BSA/PBS at the following dilutions:[80] anti-HDAg SE1679 rabbit polyclonal serum, 1/500; anti-DENV-E 3H5 mAb, 1/800; anti-NS5A 9E10 mAb, 1/1,000; and anti-HBcAg serum, 1/500. After three washes with 1% BSA/ PBS, cells were incubated for 1 h with the corresponding secondary antibodies (Molecular Probes, The Netherlands) at a 1/1000 dilution: donkey anti-rabbit Alexa Fluor 488 (Cat # A-21206); donkey anti-mouse Alexa Fluor 555 (Cat # A-31570); and goat anti-human Alexa Fluor 555 (Cat # A-21433) sera. Cells were washed three times with PBS and then stained for nuclei with Hoechst 33342 (Molecular Probes) for 5 min. After two washes in PBS, cells were imaged with an Axiovert

135 M microscope (Zeiss, Germany) equipped with a DC350FX camera (Leica, Germany), and images were analyzed with the ImageJ software (imagej.nih.gov).

**In vivo experiments**. All experiments were performed in accordance with the European Union guidelines for approval of the protocols by the local ethics committee (Authorization Agreement C2EA-15, "Comité Rhône-Alpes d'Ethique pour l'Expérimentation Animale", Lyon, France— APAFIS#1570−2015073112163780). Primary human hepatocytes (PHH, Corning, BD Gentest) were intrasplenically injected into FRG mice[40], a triple-mutant mouse knocked out for fumarylacetoacetate hydrolase ($fah^{-/-}$), recombinase-activating gene 2 ($rag2^{-/-}$), and interleukin 2 receptor gamma chain ($IL2rg^{-/-}$), 48 h after adeno-uPA conditioning[41,42]. Mice were subjected to NTBC cycling during the liver repopulation process[41]. Mice with human serum albumin (HSA) levels > 15 mg/mL, as determined using a Cobas C501 analyzer (Roche Applied Science), were inoculated with virus preparations by intraperitoneal injection. Sera were collected at different time points before and after infection. Mice were killed 10–14 weeks post infection.

**Statistical analysis**. Data are shown as means ± standard deviations. Statistical analyses were performed using two-sample Student's t tests assuming unequal variance. The p-values are represented according to the following convention: $p > 0.05$ (nonsignificant, ns); $p < 0.05$ (*); $p < 0.01$ (**).

**Reporting summary**. Further information on research design is available in the Nature Research Reporting Summary linked to this article.

## Data availability

The datasets generated during this study are available from the corresponding author upon reasonable request. The source data underlying figures and Supplementary Figures are provided as a Source Data file.

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

## Acknowledgements

We thank Massimo Levrero and Fabien Zoulim for helpful discussions and critical reading of this paper. We thank Christelle Granier and Solène Denolly and all the members of our laboratory for their support and encouragement. We are grateful to Philippe Desprès for the 3H5 DENV-E antibody, Mansun Law for the AR3A HCV-E2 antibody, and Charles Rice for the Huh-7.5 cells and the 9E10 HCV-NS5A antibody. We thank the Plateforme de Thérapie Génique in Nantes (France) for the production of the in vivo-certified lots of adeno-uPA vector. We thank Jean-François Henry, Nadine Aguilera and Jean-Louis Thoumas from the animal facility (PBES, Plateau de Biologie Experimental de la Souris, UMS3444/CNRS, US8/Inserm, ENS de Lyon, UCBL), and Anaïs Ollivier for her technical help in handling of mice. We acknowledge the contribution of the ANIRA-Genetic Analysis facility of SFR Biosciences (UMS3444/CNRS, US8/Inserm, ENS de Lyon, UCBL) for technical assistance and support. We thank Didier Décimo for support with the BSL3 facility. We thank Christelle Fabrer-Boulé from the electron microscopy studies at the "Centre Technologique des Micro-structures (CTμ)" at UCBL. We thank N. Gadot (Plateforme Anatomopathologie Recherche, Centre Léon Bérard, F-69373 Lyon, France) for the IHC analyses. This work was supported by the French "Agence Nationale de la Recherche sur le SIDA et les hépatites virales" (ANRS) and the LabEx Ecofect (ANR-11-LABX-0048) of the "Université de Lyon", within the program "Investissements d'Avenir" (ANR-11-IDEX-0007) operated by the French National Research Agency (ANR).

## Author contributions

J.P.-V. and F.-L.C. conceived the study. J.P.-V., F.A., B.B., C.M., F.F. and F.-L.C. designed and performed experiments. J.P.-V., F.A., B.B., N.F., C.S., F.F. and F.-L.C. analyzed the data. J.P.-V. and F.-L.C. wrote the paper with contributions from all authors.
