## [Peer Review File · Nature Communications]

Reviewers' Comments:

Reviewer #1:

Remarks to the Author:

NCOMMS-18-21856

Hepatitis delta virus (HDV) has long been known to co-infect people in a manner dependent on the hepatitis B virus and, indeed, was discovered based on its association with severe hepatitis in HBV patients. This dependence is due to the need for the hepadnavirus envelope to release infectious particles containing the HDV RNA and protein enveloped by the hepadnavirus surface antigen. Perez-Vargas et al describe results indicating that HDV might be able to use the envelope proteins of other viruses to produce infectious particles. The observations involve combinations of cell culture based experiments in which infectious HDV was found to be released from cells co-expressing the envelope of other viruses besides HBV, as well as experiments using mice with "humanized" livers in which HDV spread occurred via HCV rather than HBV, although at much lower levels. The authors findings not only have the potential to dramatically expand our understanding of the egress of HDV RNA-protein complexes from cells but also imply that HDV might affect the pathogenesis of numerous other viruses, as it is known to do for Hepatitis B virus. Thus, the potential impact of this study is quite large. However, given this potential impact, it is imperative that several weaknesses in the cell-culture based experiments several be addressed to convincingly demonstrate that the authors' conclusions are correct. These weaknesses (addressed specifically below), in addition to inconsistencies in some of the data and the use of somewhat unorthodox cell culture conditions undermine confidence in the robustness of the experimental results.

Major points

1. The conclusion that infectious HDV particles are secreted from cells expressing VSV or HCV glycoproteins is indeed suggested by the results in Figs. 1 and 2; however, there are concerns. The biggest is that the purported secreted viral particles are not characterized sufficiently and the characterization that is provided - immunoprecipitation - raises questions about their identity. The immunoprecipitation of the particles is very inefficient - less than 10%. This inefficiency is masked by the log representation of the graph in Fig. 1B. There is no need for a log representation in this figure; it should be changed. It seems possible that the observed release of what are presumed to be enveloped particles might be just HDV RNPs. Further analysis of the particles, in particular determining whether the RNA is genome, antigenome or some combination of both, would help to conclude one way or the other. In addition, in Fig. 1B, the authors suggest a very unusual HDAG composition for the HDV particles released from cells co-expressing different virus glycoproteins (even for regular "HDV"). Numerous studies of HDV particles obtained from cells, infected animals and patients consistently show similar amounts of S-HDAg and L-HDAg. The authors, from labeling of the immunoblot, propose that the particles consist mostly of L-HDAg, with very little S-HDAg present (except following immunoprecipitation with anti-VSV G antibody). There is no discussion of this highly unusual HDAG composition. On the other hand, without showing specific L-HDAg and S-HDAg controls on the immunoblot, it is possible the electrophoresis is not resolving the two species (that is, the labeling is incorrect) and that it is not possible to determine the relative amounts of S-HDAg and L-HDAg from the blot shown.

2. A second major concern is that the authors have not conclusively shown that the particles they have obtained are actually infectious. Several additional experiments are necessary to prove that the authors are observing actual infections rather than, for example, adherence of material to cells. A time course showing accumulation of HDV RNA and protein over several days following incubation would be more convincing than a single time point on day 7. Also more convincing would be demonstration of

antigenomic RNA accumulation (assuming that the released RNA is genome) and an increase with time in the amount of L-HDAg, which only occurs during replication.

3. The authors included 2% DMSO in the cell cultures used to produce HDV particles and for infection experiments. The stated purpose was to retard cell growth. However, this treatment is unnecessary (2% DMSO is not typically included in experiments used to produce HDV nor to analyze infection in Huh 106 cells) and its use here raises questions about the generality of the results. Perhaps 2% DMSO is somewhat toxic (it does retard cell growth) and leads to the release of HDV from cells stressed by expression of certain viral glycoproteins. At least some of the production and infection experiments should be repeated in the absence of DMSO.

Additional points

1. There was considerable variability between some similar experiments, or unexplained inconsistencies. In Fig. 3, for example, the levels of HDV RNA detected in mock infected cells varied by almost 10-fold between panel A and panel B. While the RNA analyses suggest that HDV obtained from cells expressing VSV G replicates to 10-fold higher levels than HDV with an HBV envelope (Fig 2A), the immunofluorescence analysis shows approximately 5 times more cells positive for HDV with the HBV envelope (Fig. 2C).
2. Page 3, second paragraph, second sentence. The statement that HDV does not meet the criteria for the definition of a virus is overstated. HDV is a satellite virus, with HBV as the only known helper.
3. Figure 5. The legend describes black and gray bars, there are only solid black and hashed bars.
4. Supplemental Fig. 1; Group 9. According to the table in Fig. 6, only 4 animals were in this group, yet there are about 9 lines on the graph. There is no indication of what the colors represent in this graph nor on any of the others in this figure.

Reviewer #2:

Remarks to the Author:

Perez-Vargas et al. report that hepatitis D virus (HDV), which was thought to be a cognate satellite virus of hepatitis B Virus (HBV), may be enveloped by the surface glycoproteins (GPs) from a variety of viruses for its assembly and the release. The authors also found that the production and infection of HDV is not restricted to liver cells, but is restricted by the GPs enveloping HDV and the corresponding receptors expressed on cells. The authors performed various cell culture and animal experiments to support their observations. Co-expression of HDV RNPs and individual viral GP (HBV, VSV, HCV, etc) in hepatoma cell lines led to release of infectious HDV particles. The infection of different GP-enveloped HDV could be blocked by antibodies to the corresponding viral envelop proteins. Notably, in both HBV or HCV infected mice HDV viremia was detected, confirming the observations in a physiologically relevant experimental system.

Based on these lines of evidence, the authors propose that HDV may have an origin independent of HBV, and could potentially be propagated by viruses other than HBV. The authors observation is surprising, given the scarcity of literature hinting HDV infection in the absence of an HBV infection. The authors have carved a unique angle to investigate the HDV propagation and provided both cell culture and animal evidences to support their hypothesis, which if proven true will enrich our understanding of HDV origin and the interplay of different viruses. However, several issues need to be addressed to further validate their hypothesis.

Major issues:

1. In figure 1 and figure 4, the key plasmid pSVLD3 was first developed in the Taylor lab (Kuo, et al, 1989, J. Virol) as a trimer of HDV genome and used for HDV replication. However, as it was indicated later (Taylor, 2006, Curr Top Microbiol Immunol), that the unit-length genomic or antigenomic HDV

RNA was mostly DNA-directed rather than RNA-directed, and since this does not recapitulate the authentic HDV replication, an improved construct was made to have slightly larger than one unit length to produce RNA-derived transcripts (Lazinski & Taylor, 1994, J. Virol). The data produced in Figure 1 and 4 may not hold if the production of HDV RNPs is formed in way that deviates from the authentic mechanism. Two methods could be used: 1) NTCP expressing cells/cell lines expressing various viral GPs is infected by recombinant HBV enveloped HDV virus to test if infectious HDV particles is secreted, and 2) use the improved constructs to validate the findings.

2. In figure 1a, the release of virus through cell death should be ruled out.

Minor issues:

1. Figure 1b, the VSV- Δ p 41A1 has a different pattern, e.g, much more s-HDAg and a 55 kDa band. A discussion of the difference, cause, and implication would be helpful.

2. Figure 2c should include light-field or nuclei staining.

3. Figure 5, it is not clear what is the HDV-RNA expressing cell, what construct is used to drive HDV-RNA expression?

4. Is there any explanation why the HDV infection is not reported to be dependent on HCV, if HCV propagates HDV indeed rather efficiently?

5. At numerous occasions the authors refer to "data not shown". This reviewer regards it as important to show all data relevant to the study in the manuscript

6. Page 4: "While HDV was expressed..." rephrase as RNA is not expressed but transcribed or in this case replicated. Proteins are expressed.

7. Page 8, line: please correct/spell out the mutant alleles in the FRG mice: fumarylacetoacetate hydrolase (fah-/-), recombinase activating gene 2 (rag2-/-), interleukin 2 receptor gamma chain (IL2RgNULL)

8. Page 12: subheading, delta particle NOT particles

Reviewer #3:

Remarks to the Author:

The origin of HDV attracts many interesting speculations. Current study proposed it coming from cellular circular RNAs that captured by HBV envelopes, so it is possible other viruses also enabling the packaging of naked HDV RNPs. They tested several viruses, such as HCV, dengue, HIV, et al, and showed HCV, denge and WNV, among others, could package the HDV RNPs into their envelope proteins and pass the HDV into next round of infections in cell cultures or in human hepatocyte chimera mice.

Though the results appeared to be interesting, their validation of pseudo-typed HDV and its infections is incomplete and many basic HDV RNA and proteins analysis are missing.

The comments are for authors' reference.

1. The claimed of packaging of HDV RNP by HBV Surface proteins, or by VSV or HCV GPs are interesting by in vitro co-transfection. This finding is supported by their immune-precipitation of these

pseudo-typed HDVs, using anti-HBs or anti-VSV or anti-HCV GP. However, the validation of these packaged HDV falls far behind. To confirm the packaged HDV RNP, the authors only showed an ambiguous detection of so-called HDV large HDAg, but not HDV RNA. It is essentially to include a HDV virus packaged by HBV surface protein as a positive control. Such virion has to contain the HDV small delta antigen, and genomic HDV RNA, other than the so-called large HDAg. A Northern blot to confirm an intact, full-sized HDV RNA is required. The RT-PCR quantitation cannot distinguish the viral genomic vs. antigenomic HDV RNAs, either about the size. (In fact, the large HDAg detected in western blot appeared to be suspicious. Other HDAg-specific antibody is required, as it is difficult to understand why no small HDAg is co-packaged).

2. The packaging of HDV RNP by HBsAg required specific isoprenylation of large HDAg. Do the rescue of HDV RNPs require the same modification or not? This can be easily studied by using isoprenylation inhibitor currently available.

3. The authors tried to band the VSV or HCV GP-packaged HDV virions by CsCL gradient analysis. They succeeded in identifying the putative pseudo-typed HDV in unique density fractions. Again, their only data based upon RT-PCR assay for HDV RNA. Northern and western blots to show HDV genomic RNA and both large and small delta antigens are essential. Finally, as the HDV virions are so abundant, it is necessary to do a simple EM study for these fractions to visualize the size, distribution of these pseudo-type HDV particles.

4. In their co-infection experiments, though HCV or other viruses appeared able to rescue the intracellular HDV RNPs and resulted in efficient next round infections, the data are not comprehensive. The authors relied only HDV RNA quantification by RT-PCR, however, they failed to provide either northern blot or western blot to show the simultaneous presence of HDV RNA or delta antigens. These are easily to show, as the HDV RNA titers are so high by their data. Besides, it is important to document the co-presence of HDAg and HCV antigen or dengue virus antigen in the same human hepatocytes from the chimera mice. Without these collaborating data, the HDV RNA RT-PCR seems shaky. It should be pointed that currently there is no approved HDV RNA assays, and many in-house assays suffer from varying or inconsistent performance.

5. Finally, the HCV or dengue virus infections in humanized chimera mice took a lot of effort and showed intriguing results. Other than insufficient virological data as mentioned in point 3, the authors may need to study the natural HCV/HDV coinfection in human intravenous drug abusers who frequently co-infected by HBV/HDV/HCV. Do these patients carry HDV RNA within HCV envelope?

Dr. François-Loïc Cosset
Scientific and Executive Director

International Center for Infectology Research
21 Avenue Tony Garnier
69365 LYON, France
<http://ciri.inserm.fr>

December 2nd, 2018

Dear Reviewers

Please find enclosed our revised manuscript entitled “Enveloped viruses distinct from HBV induce dissemination of hepatitis D virus *in vivo*” by Jimena Perez-Vargas, Fouzia Amirache, Bertrand Boson, Chloé Mialon, Camille Sureau, Floriane Fusil, and myself, which we would like to publish in *Nature Communications* as a Research Article.

We greatly appreciated the Editor and Reviewers’ helpful and constructive comments, which we have all taken into account to improve our manuscript by performing additional experiments. Overall, we believe that we have succeeded to provide a more detailed description of these novel HDV particles and to clarify most issues raised by all Reviewers in this revised version of our manuscript.

We have revised the manuscript accordingly (see manuscript copy with changes underlined) and we provide a point-by-point response to these comments below (in blue).

The additional results, as per Reviewers’ requests, are:

- Northern blot analysis of HDV particles produced with unconventional GPs (Figure 1C) and of cells infected with these particles (Figure 3D).
- Determination by strand-specific RT-PCR of HDV RNA unit size in HDV particles produced with unconventional GPs (Figure 1B, Figure 1H), with live viruses (Figure 6), and in sera of co-infected animals (Supplemental Figure 7).
- Western blot analysis of HDV particles produced with HBV, VSV and HCV glycoproteins (Figure 1D) and of cells infected with these particles (Figure 3E).
- Electron microscopy analysis of HDV particles produced with HBV, VSV and HCV glycoproteins (Figure 1F and Supplemental Figure 3).
- Quantitative analysis of HDV intracellular and extracellular RNAs by strand-specific RTqPCR assays that detect genomic and antigenomic RNAs (Figure 3 and Supplemental Figure 1).
- Use of Lonafarnib in production (and subsequent infection) experiments (Figure 3A, B).
- Time-course analysis of genomic vs. antigenomic HDV RNAs and of L-HDAg and S-HDAg forms in infected cells (Figure 3C, D).
- Assessment of co-infection of cells by HDV and live helper viruses (HBV, HCV and DENV) by immuno-fluorescence assays (Figure 5D and Supplemental Figure 5).
- Demonstration that HDV particles can be formed with non-HBV glycoproteins both *via* transfection of pSVLD3 plasmid and *via* infection with “helper-free” HDV particles in GP-expressing cells (Supplemental Figure 6).

The other important changes in our revised manuscript are:

- Statistical analysis of the data.
- Assessment of production of HDV particles by HDV/DENV-co-infected mosquito cells (Supplemental Figure 6).
- Demonstration that identical production of HDV particles can be achieved from cells cultured in media containing, or not, 2% DMSO (Supplemental Figure 2).
- Results of co-infection by HDV and HCV in a second cohort of human liver mice (N=24; Supplemental Figure 8).

- Reorganization of the manuscript text, figures and supplementary figures to address, on the whole, all comments of the Reviewers.

We thank you very much for your interest and time in considering our revised manuscript for publication in *Nature Communications*.

Sincerely yours,

Dr. FL Cosset

Point-by-point Reply (see literature cited at the end of this document)

Reviewer #1 (Remarks to the Author):

Hepatitis delta virus (HDV) has long been known to co-infect people in a manner dependent on the hepatitis B virus and, indeed, was discovered based on its association with severe hepatitis in HBV patients. This dependence is due to the need for the hepadnavirus envelope to release infectious particles containing the HDV RNA and protein enveloped by the hepadnavirus surface antigen. Perez-Vargas et al describe results indicating that HDV might be able to use the envelope proteins of other viruses to produce infectious particles. The observations involve combinations of cell culture based experiments in which infectious HDV was found to be released from cells co-expressing the envelope of other viruses besides HBV, as well as experiments using mice with “humanized” livers in which HDV spread occurred via HCV rather than HBV, although at much lower levels. The authors findings not only have the potential to dramatically expand our understanding of the egress of HDV RNA-protein complexes from cells but also imply that HDV might affect the pathogenesis of numerous other viruses, as it is known to do for Hepatitis B virus. Thus, the potential impact of this study is quite large. However, given this potential impact, it is imperative that several weaknesses in the cell-culture based experiments several be addressed to convincingly demonstrate that the authors’ conclusions are correct. These weaknesses (addressed specifically below), in addition to inconsistencies in some of the data and the use of somewhat unorthodox cell culture conditions undermine confidence in the robustness of the experimental results.

Major points

1. The conclusion that infectious HDV particles are secreted from cells expressing VSV or HCV glycoproteins is indeed suggested by the results in Figs. 1 and 2; however, there are concerns. The biggest is that the purported secreted viral particles are not characterized sufficiently and the characterization that is provided - immunoprecipitation – raises questions about their identity. The immunoprecipitation of the particles is very inefficient – less than 10%. This inefficiency is masked by the log representation of the graph in Fig. 1B. There is no need for a log representation in this figure; it should be changed. It seems possible that the observed release of what are presumed to be enveloped particles might be just HDV RNPs. Further analysis of the particles, in particular determining whether the RNA is genome, antigenome or some combination of both, would help to conclude one way or the other. In addition, in Fig. 1B, the authors suggest a very unusual HDAg composition for the HDV particles released from cells co-expressing different virus glycoproteins (even for regular “HDV”). Numerous studies of HDV particles obtained from cells, infected animals and patients consistently show similar amounts of S-HDAg and L-HDAg. The authors, from labeling of the immunoblot, propose that the particles consist mostly of L-HDAg, with very little S-HDAg present (except following immunoprecipitation with anti-VSV G antibody). There is no discussion of this highly unusual HDAg composition. On the other hand, without showing specific L-HDAg and S-HDAg controls on the immunoblot, it is possible the electrophoresis is not resolving the two species (that is, the labeling is incorrect) and that it is not possible to determine the relative amounts of S-HDAg and L-HDAg from the blot shown.

Reply: We have addressed these points and we are happy to provide a more detailed characterization of VSV- Δp and HCV- Δp particles, as described below :

1 - Demonstration that HDV RNAs in particles are genomes rather than antigenomes. We used a strand-specific RTqPCR assay (Li et al., 2006) to quantify HDV genomic RNA (gRNA) and antigenomic RNA (agRNA) in lysates and supernatants of transfected and/or infected cells (see new Supplemental Figure 1A). The enrichment of HDV gRNAs (panel A) in secreted particles is reflected by the gRNA/agRNA ratios (panel C), which were up to 800-fold higher in HDV, VSV- Δp or HCV- Δp particles than in the lysates of their corresponding producer cells. As shown in panels B and E, we noted no significant increase over time post-transfection of the low amounts of HDV agRNAs detected in the supernatants, in sharp contrast to the extracellular HDV gRNAs that increased by up to 1,000-fold (panels A and D). Owing to the high sensitivity of the RTqPCR assay, these low levels HDV agRNAs could be due to some background of cell death induced by the combination of GP transfection and extended culture conditions of these cells (up to 9 days). Note that identical extracellular HDV agRNAs levels were detected for VSV- Δp and HCV- Δp particles as compared to “normal” HDV particles (*i.e.*, with HBV GPs). We believe that these new results show that HDV particles generated with unconventional GP incorporate full-length genomic RNA.

2 - Improvement regarding the detection of HDAg species present in viral particles. We replaced the HDAg co-IP analysis, which was not possible to improve at this stage, by a Western blot analysis of

the different types of HDV particles that were purified by ultracentrifugation on a sucrose cushion. The results clearly show that both L-HDAg and S-HDAg are incorporated at similar levels and ratios in the purified viral particles generated with HCV and VSV GPs as compared to “normal” HDV particles produced with HBV GPs.

3 – Concerning the RNA colP that precipitated 5 to 11% of HDV RNAs, we do not expect that the efficiency could be higher because of the competition exerted by the SVPs for either type of particles that outnumber the infectious particles. We respectfully request to keep the log representation of the graph in Figure 1E, in order to better show the results of the Flow through RTqPCR values.

2. A second major concern is that the authors have not conclusively shown that the particles they have obtained are actually infectious. Several additional experiments are necessary to prove that the authors are observing actual infections rather than, for example, adherence of material to cells. A time course showing accumulation of HDV RNA and protein over several days following incubation would be more convincing than a single time point on day 7. Also more convincing would be demonstration of antigenomic RNA accumulation (assuming that the released RNA is genome) and an increase with time in the amount of L-HDAg, which only occurs during replication.

Reply: We have performed all these experiments. The panels C-E of the new Figure 3 provides a time course analysis in infected cells over several days following inoculation. Using a strand-specific RTqPCR assay for HDV RNA, we show that not only genomic HDV RNAs but also antigenomic RNAs are amplified from day 3 to day 9 post-infection and accumulate in infected cells (panel C). Likewise, we confirm by Northern blot analysis of these infected cells that HDV RNAs accumulate in these cells (Panel D). Finally, we show that HDAg protein levels also increase with a progressive appearance of L-HDAg, which marks productive infection (panel E).

In addition to the other pieces of evidence such as i) inoculation of cell expressing vs. not expressing the receptors (now Figure 4A), ii) co-infection and transmission assays with live helper viruses (now Figure 6 and Supplemental Figure 6) and iii) propagation in experimentally infected animals (now Figure 7 and Supplemental Figures 7 & 8), we believe that altogether, these results convincingly show that the particles are infectious.

3. The authors included 2% DMSO in the cell cultures used to produce HDV particles and for infection experiments. The stated purpose was to retard cell growth. However, this treatment is unnecessary (2% DMSO is not typically included in experiments used to produce HDV nor to analyze infection in Huh 106 cells) and its use here raises questions about the generality of the results. Perhaps 2% DMSO is somewhat toxic (it does retard cell growth) and leads to the release of HDV from cells stressed by expression of certain viral glycoproteins. At least some of the production and infection experiments should be repeated in the absence of DMSO.

Reply: We used Williams E-based medium for production or infection with HDV particles in Huh-7-derived cells (including Huh-106 and Huh-7.5 cells). While this Reviewer is correct to say that DMSO-containing medium is not typically included in such experiments, we supplemented our medium with 2% DMSO in this study since, when we started the project, the current procedures to grow hepatocyte-derived cell lines for several days recommended such media both to maintain cell differentiation (Bauhofer et al., 2012, Sainz & Chisari, 2006) and to induce or maintain NTCP expression (Watashi et al., 2014, Yan et al., 2012). For example, it was shown that DMSO-containing media strongly increase HDV and HBV infection efficacy in HepG2^{NTCP} and Huh-7^{NTCP} cells, as compared to DMSO-free media (Iwamoto et al., 2014, Ni et al., 2014).

Importantly, as requested by this Reviewer, we now show in the new Supplemental Figure 2, first, that 2% DMSO does not induce more cell toxicity as compared to cell cultures grown in DMSO-free medium (panel A) and, second, that omitting DMSO does not change the production levels and infectivity of HDV, VSV-Δp and HCV-Δp particles (panels B and C).

Additional points

1. There was considerable variability between some similar experiments, or unexplained inconsistencies. In Fig. 3, for example, the levels of HDV RNA detected in mock infected cells varied by almost 10-fold between panel A and panel B. While the RNA analyses suggest that HDV obtained from cells expressing VSV G replicates to 10-fold higher levels than HDV with an HBV envelope (Fig 2A), the immunofluorescence analysis shows approximately 5 times more cells positive for HDV with the HBV envelope (Fig. 2C).

Reply: The previous immunofluorescence analysis of Figure 2C was displayed to provide a qualitative assessment of HDAg nuclear localization upon infection. As requested by this Reviewer, we have

repeated the experiment and we provide in the revised Figure 2B images that are more consistent with the quantification of infectivity (now shown in Figure 2C) and that include nuclei staining with Hoechst. As for the variability between panel A and panel B of previous Figure 3 (now Figure 4), *i.e.*, there is 3- to 5-fold differences in the HDV RNA RTqPCR results obtained for the mock-treated supernatants for panel A vs. panel B. This is explained by the fact that either experiment type (*i.e.*, neutralization in panel A and receptor blocking in panel B) was performed at different sessions of the study, which induces experimental variations. Yet, we had made sure that the sizes of the HDV, VSV- Δ p and HCV- Δ p particles inputs in either panel A or panel B were identical in order to provide accurate comparisons between conditions in each panel.

2. Page 3, second paragraph, second sentence. The statement that HDV does not meet the criteria for the definition of a virus is overstated. HDV is a satellite virus, with HBV as the only known helper.

Reply: We have removed this sentence as requested.

3. Figure 5. The legend describes black and gray bars, there are only solid black and hashed bars.

Reply: We have corrected the legend in this Figure (now revised Figure 6).

4. Supplemental Fig. 1; Group 9. According to the table in Fig. 6, only 4 animals were in this group, yet there are about 9 lines on the graph. There is no indication of what the colors represent in this graph nor on any of the others in this figure.

Reply: We thank this Reviewer for pointing this. Like for the “Mocks” group (Group #10), but also in other groups (not shown, for sake of clarity), we tested the samples for all three viruses (HBV, HCV and HDV) in Group #9, whose mice were infected with HDV only, and this is now indicated in the revised legend with the indicated color codes (blue: HBV; black: HCV; red: HDV). Note that one animal died after week 8 (P4), which explains that there are 9 lines after P4 (12 lines until P4) in this graph.

Reviewer #2 (Remarks to the Author):

Perez-Vargas et al. report that hepatitis D virus (HDV), which was thought to be a cognate satellite virus of hepatitis B Virus (HBV), may be enveloped by the surface glycoproteins (GPs) from a variety of viruses for its assembly and the release. The authors also found that the production and infection of HDV is not restricted to liver cells, but is restricted by the GPs enveloping HDV and the corresponding receptors expressed on cells. The authors performed various cell culture and animal experiments to support their observations. Co-expression of HDV RNPs and individual viral GP (HBV, VSV, HCV, etc) in hepatoma cell lines led to release of infectious HDV particles. The infection of different GP-enveloped HDV could be blocked by antibodies to the corresponding viral envelop proteins. Notably, in both HBV or HCV infected mice HDV viremia was detected, confirming the observations in a physiologically relevant experimental system.

Based on these lines of evidence, the authors propose that HDV may have an origin independent of HBV, and could potentially be propagated by viruses other than HBV. The authors observation is surprising, given the scarcity of literature hinting HDV infection in the absence of an HBV infection. The authors have carved a unique angle to investigate the HDV propagation and provided both cell culture and animal evidences to support their hypothesis, which if proven true will enrich our understanding of HDV origin and the interplay of different viruses. However, several issues need to be addressed to further validate their hypothesis.

Major issues:

1. In figure 1 and figure 4, the key plasmid pSVLD3 was first developed in the Taylor lab (Kuo, et al, 1989, J. Virol) as a trimer of HDV genome and used for HDV replication. However, as it was indicated later (Taylor, 2006, Curr Top Microbiol Immunol), that the unit-length genomic or antigenomic HDV RNA was mostly DNA-directed rather than RNA-directed, and since this does not recapitulate the authentic HDV replication, an improved construct was made to have slightly larger than one unit length to produce RNA-derived transcripts (Lazinski & Taylor, 1994, J. Virol). The data produced in Figure 1 and 4 may not hold if the production of HDV RNPs is formed in way that deviates from the authentic mechanism. Two methods could be used: 1) NTCP expressing cells/cell lines expressing various viral GPs is infected by recombinant HBV enveloped HDV virus to test if infectious HDV particles is secreted, and 2) use the improved constructs to validate the findings.

Reply: We thank this Reviewer for pointing out the original reference to this key construct, which is duly cited now (Kuo et al., 1989). Regarding his/her specific point, we had used in Figure 6 a variation of the

method #1 he/she suggested. Accordingly, following infection of naïve cells with VSV- Δ p particles (*i.e.*, “helper-free” VSV-G enveloped HDV particles), the cells were superinfected with live HCV, HBV or DENV viruses. The results show that we could rescue infectious HDV particles, indicating that both pSVLD3 transfection and “helper-free” HDV infection (such as VSV- Δ p) processes leads to expression of HDV RNA that can be transmitted as infectious particles. Note that a similar conclusion can be deduced from the mouse infection data since these animals were inoculated with “helper-free” HBsAg-enveloped HDV virus before, after or concomitantly with live HCV or HBV (Figure 7).

Finally, we show in the new Supplemental Figure 6 the results of transmission experiments using both Huh-7.5 human hepatoma and C6/36 mosquito cells that were infected with supernatants from HDV/DENV co-infected cells. We found that these secondary HDV/DENV-infected Huh-7.5 and C6/36 cells could replicate, assemble and transmit infectious HDV particles to tertiary cells.

In our opinion, these results also demonstrate that HDV RNA can be transmitted *via* processes that involve authentic HDV replication.

2. In figure 1a, the release of virus through cell death should be ruled out.

Reply: We now provide indirect evidence to address this point, which is very difficult to formally rule out since most of the GPs studied here can intrinsically cause cell death (albeit through different pathways). We evaluated the cytotoxicity in transfected cells at different time points of collection of HDV, VSV- Δ p, and HCV- Δ p particles as well as in No GP (*i.e.*, pSVLD3-transfected cells) and non-transfected control cells. Using the Pierce Cytotoxicity Assay Kit, we obtained similar levels of LDH release for both VSV- Δ p or HCV- Δ p particles and “normal” HDV particles but also for the No GP control and the non-transfected condition as shown at day 6 post-transfection in the new Supplemental Figure 2. Thus, we conclude that while the combination of long-term culture and transfection procedure is somehow harmful to cells, the release of particles may not occur through cell death first, since otherwise, the No GP control (transfected with pSVLD3 only) would induce secretion of HDV RNAs from these cells, and second, since one may conclude that “classical” HDV particles would also be released through cell death.

Note that slightly increased cytotoxicity levels were obtained when producing VSV- Δ p particles; this was expected owing to the previously known fusogenic activity of VSV-G (see *e.g.*, (Arai et al., 1998) for VSV-G-pseudotyped lentiviral vectors), which, ultimately, does not preclude production of such vectors.

Minor issues:

1. Figure 1b, the VSV- Δ p 41A1 has a different pattern, *e.g.*, much more s-HDAg and a 55 kDa band. A discussion of the difference, cause, and implication would be helpful.

Reply: We agree with this issue of this Reviewer, which was also pointed out by the two other Reviewers. Accordingly, we have replaced this HDAg co-IP analysis, which was difficult to improve, by a Western blot analysis of the different types of HDV particles that were purified by ultracentrifugation on a sucrose cushion. The results unambiguously show that both L-HDAg and S-HDAg are incorporated at similar levels and ratios in the purified viral particles generated with HCV and VSV GPs as compared to “normal” HDV particles produced with HBV GPs.

2. Figure 2c should include light-field or nuclei staining.

Reply: As requested by this Reviewer, we have repeated the experiment and we provide in the revised Figure 2B images that show nuclei staining (Hoechst) in inoculated cells.

3. Figure 5, it is not clear what is the HDV-RNA expressing cell, what construct is used to drive HDV-RNA expression?

Reply: We infected these cells with VSV-G-coated HDV particles (VSV- Δ p). Cells were then superinfected with the indicated live viruses (see our reply to Major issue #1 of this Reviewer).

4. Is there any explanation why the HDV infection is not reported to be dependent on HCV, if HCV propagates HDV indeed rather efficiently?

Reply: We are deeply interested by finding an explanation to this question, which is also raised by Reviewer #3. Indeed, one of our future plans is to attempt detection of HCV-dependent HDV propagation in selected patient cohorts, which is difficult clinically and logistically. As for tentative explanation, it is possible that direct or indirect (*e.g.*, immune-mediated) interference mechanisms may impede in the long term HDV/HCV co-infections *in vivo*, though they may occur in different contexts. For example, HDV-induced activation of the innate immune response, which is known to have little/no effect on HDV itself (Alfaiate et al., 2016, Zhang et al., 2018), may impact several markers of co-infection, such as for HCV that is interferon-sensitive in contrast to HBV (Lutgehetmann et al., 2011, Mutz et al., 2018).

Thus, overall, we propose that what might determine eventual successful vs. sporadic transmission and propagation of HDV with non-HBV helper viruses would reside in the balance between biochemical/virological HDV RNP compatibility with the GP of these helper viruses vs. potential (immunological) mechanisms of interference, though the immune status of individuals, such as immune-suppression, may favor transmission of such HDV co-infections. This putative explanation is included in the revised Discussion.

5. At numerous occasions the authors refer to “data not shown”. This reviewer regards it as important to show all data relevant to the study in the manuscript

Reply: We have complied with the request of this Reviewer and we now display these data not shown in Figure 3 (kinetics of HDV replication in infected cells), in Figure 5 (results of infectivity for all HDV/GP combinations), in Supplemental Figure 5 (assessment of co-infection by IF), in Supplemental Figure 6 (infection of mosquito cells), and in Supplemental Figure 8 (results of infection from a second cohort of HuHep mice). The only results that are not shown are the FAH staining of humanized livers, as they have been published previously by us (Calattini et al., 2015) and others (Bissig et al., 2010).

6. Page 4: “While HDV was expressed...” rephrase as RNA is not expressed but transcribed or in this case replicated. Proteins are expressed.

Reply: We have changed this sentence to “While HDV RNA accumulated...”

7. Page 8, line: please correct/spell out the mutant alleles in the FRG mice: fumarylacetoacetate hydrolase (fah^{-/-}), recombinase activating gene 2 (rag2^{-/-}), interleukin 2 receptor gamma chain (IL2Rg^{NULL})

Reply: We have also modified this sentence accordingly (revised Material and Methods).

8. Page 12: subheading, delta particle NOT particles

Reply: We have introduced the requested change in this subheading.

Reviewer #3 (Remarks to the Author):

The origin of HDV attracts many interesting speculations. Current study proposed it coming from cellular circular RNAs that captured by HBV envelopes, so it is possible other viruses also enabling the packaging of naked HDV RNPs. They tested several viruses, such as HCV, dengue, HIV, et al, and showed HCV, dengue and WNV, among others, could package the HDV RNPs into their envelope proteins and pass the HDV into next round of infections in cell cultures or in human hepatocyte chimera mice.

Though the results appeared to be interesting, their validation of pseudo-typed HDV and its infections is incomplete and many basic HDV RNA and proteins analysis are missing.

The comments are for authors' reference.

1. The claimed of packaging of HDV RNP by HBV Surface proteins, or by VSV or HCV GPs are interesting by in vitro co-transfection. This finding is supported by their immune-precipitation of these pseudo-typed HDVs, using anti-HBs or anti-VSV or anti-HCV GP. However, the validation of these packaged HDV falls far behind. To confirm the packaged HDV RNP, the authors only showed an ambiguous detection of so-called HDV large HDAg, but not HDV RNA. It is essentially to include a HDV virus packaged by HBV surface protein as a positive control. Such virion has to contain the HDV small delta antigen, and genomic HDV RNA, other than the so-called large HDAg. A Northern blot to confirm an intact, full-sized HDV RNA is required. The RT-PCR quantitation cannot distinguish the viral genomic vs. antigenomic HDV RNAs, either about the size. (In fact, the large HDAg detected in western blot appeared to be suspicious. Other HDAg-specific antibody is required, as it is difficult to understand why no small HDAg is co-packaged).

Reply: We have performed these experiments and we provide in this revised manuscript a more detailed characterization of VSV- Δ p and HCV- Δ p particles :

1 – Incorporation of genomic HDV RNA in viral particles.

First, we provide in revised Figure 1C a result of Northern blot that confirm the presence of an intact, full-sized HDV RNA in pellets of particles purified by ultracentrifugation on a 30% sucrose cushion.

Second, we used a strand-specific RTqPCR assay (see revised Material and Methods) to quantify HDV genomic RNA (gRNA) and antigenomic RNA (agRNA) in lysates and supernatants of transfected and/or

infected cells (see new Supplemental Figure 1A). The enrichment of HDV gRNAs (panel A) in secreted particles is reflected by the gRNA/agRNA ratios (panel C), which were up to 800-fold higher in HDV, VSV- Δ p or HCV- Δ p particles than in the lysates of their corresponding producer cells.

Third, we designed a RT-PCR strand-specific “banding” assays with primers that allow amplification of the HDV genomic RNA. We found that HDV particles contains HDV RNA at the expected size of 1.7 kb, whether they were produced by transfection with pSVLD3 and GP-expression plasmids (Figure 1B, 1H) or by co-infection with live HCV, HBV or DENV *in vitro* (Figure 6) and in experimentally-infected animals (Supplemental Figure 7).

Fourth, showing that VSV- Δ p and HCV- Δ p form particles, we performed Electron Microscopy analysis of particles (Figure 1C and Supplemental Figure 3), which were obtained from cell supernatants purified with heparin beads, as discussed below in Point #3.

2 - Detection of HDAg species present in viral particles. We replaced the HDAg co-IP analysis, which was not possible to improve, by a Western blot analysis of the different types of HDV particles that were pelleted by ultracentrifugation on a sucrose cushion. The results (Figure 1D) clearly show that both L-HDAg and S-HDAg are incorporated at similar levels and ratios in the purified viral particles generated with HCV and VSV GPs as compared to “normal” HDV particles produced with HBV GPs.

2. The packaging of HDV RNP by HBsAg required specific isoprenylation of large HDAg. Do the rescue of HDV RNPs require the same modification or not ? This can be easily studied by using isoprenylation inhibitor currently available.

Reply: We have performed the requested experiment and we show, in Figure 3A & B, that Lonafarnib, an isoprenylation inhibitor that prevents HDV assembly and secretion (Bordier et al., 2003), could readily inhibit production and hence, transmission and replication of HDV RNA from HDV, VSV- Δ p and HCV- Δ p particles, suggesting a shared pathway of the early assembly process leading to production of all HDV particle types.

3. The authors tried to band the VSV or HCV GP-packaged HDV virions by CsCL gradient analysis. They succeeded in identifying the putative pseudo-typed HDV in unique density fractions. Again, their only data based upon RT-PCR assay for HDV RNA. Northern and western blots to show HDV genomic RNA and both large and small delta antigens are essential. Finally, as the HDV virions are so abundant, it is necessary to do a simple EM study for these fractions to visualize the size, distribution of these pseudo-type HDV particles.

Reply: We performed the requested experiment but failed to detect HDV RNA by Northern blot in that specific case, owing to their insufficient concentrations in fractions from density gradients. Indeed, the HDV RNA copy number required to perform Northern blots ($>10^7$ copies/lane – *i.e.*, 20 μ l loaded – are needed) allowed detection of full-sized HDV RNA in the 100-fold pelleted viral particles shown in Figure 1C but not in the fractions from iodixanol density gradients, owing to dilution of the sample in the gradient. Note that the aim of this latter experiment was to performed density gradient analysis from unprocessed, crude supernatants (in order to maintain native state of the sample) and that these supernatants contain *ca.* 4×10^7 copies /mL for HCV- Δ p particles (thus, the most HDV RNA-enriched fraction contain less than 2×10^5 copies/20 μ l, which is below the threshold level). As for detection of L-HDAg and S-HDAg, while we concentrated the fractions with methanol/acetone, the signals were not of sufficient quality for being displayed in Figure 1 owing to BSA levels that interfered with migration in SDS-Page.

To overcome these technical issues and address the request of this Reviewer regarding the packaging of full-sized HDV RNA, we used the above-mentioned RT-PCR banding assays and we show that the RTqPCR-positive fractions contain HDV RNA at the expected genomic size of 1.7 kb (Figure 1G).

As for EM studies, we provide in Figure 1F and Supplemental Figure 3 images of particles which were obtained from cell supernatants purified with heparin beads. We observed two types of spheres with diameters of 35-40 and 25-30 nm (Gudima et al., 2007). The small spheres likely corresponded to sub-viral particles since they were also detected when VSV-G and HCV-E1E2 were expressed alone, similar to HBV GPs (Supplemental Figure 3C & D) whereas the large spheres, that were only detected when HDV RNA were co-expressed with either GP (Supplemental Figure 3A & B), could correspond to VSV- Δ p and HCV- Δ p particles. While the concentration of these particles appeared insufficient to allow simple EM studies from the fractions of density-gradients, we believe that this consolidate the characterization of these novel HDV particles.

4. In their co-infection experiments, though HCV or other viruses appeared able to rescue the intracellular HDV RNPs and resulted in efficient next round infections, the data are not comprehensive. The authors relied only HDV RNA quantification by RT-PCR, however, they failed to provide either

northern blot or western blot to show the simultaneous presence of HDV RNA or delta antigens. These are easily to show, as the HDV RNA titers are so high by their data. Besides, it is important to document the co-presence of HDAg and HCV antigen or dengue virus antigen in the same human hepatocytes from the chimera mice. Without these collaborating data, the HDV RNA RT-PCR seems shaky. It should be pointed that currently there is no approved HDV RNA assays, and many in-house assays suffer from varying or inconsistent performance.

Reply: We show in the revised set of figures a more detailed characterization of these co-infection experiments.

First, regarding the presence of full-sized HDV RNA, using the above-mentioned RT-PCR banding assays, we show that cells co-infected with HDV and HCV or HBV or DENV express and secrete RNAs at the expected genomic size of 1.7 kb, which matches the detection of these RNAs by RTqPCR.

Second, we performed an immunofluorescence (IF) analysis of these co-infected cells to document the co-presence of HDAg with HCV, HBV or DENV antigens in the same human hepatocytes. As shown in the revised Figure 6 and new Supplemental Figure 5, in addition to cells that were mono-infected by either virus type, we could readily detect cells co-expressing the antigens of either virus combinations (HDAg with HBV core, HDAg with HCV NS5A or HDAg with DENV E), which indicates that cells were co-infected by HDV and HCV, HBV or DENV.

Third, regarding the chimera mice, we performed an IF analysis of liver sections from the co-infected animals. While we could readily detect mono- and co-infected hepatocytes in HBV/HDV co-infected mice, as shown previously by others (Lutgehetmann et al., 2012), the IF analysis of HDV/HCV infected animals was more difficult (see Figure below) and displayed rare co-infected cells with dull HDAg immunofluorescence, which may be explained at this stage by the following reasons. First, as the levels of HCV RNAs in this human liver mouse model are typically *ca.* 10-30 fold less elevated than for HBV, the propagation of HDV is less favorable with HCV helper virus than with HBV and results in a smaller proportion of infected hepatocytes. Second, at the time these animals were sacrificed (*i.e.*, at week 14 post-infection (Figure 7A)), the levels of HDV RNAs had decreased by over two-logs as compared to previous time-points (see *e.g.*, week 8 for Group#8 in Figure 7), which made the analysis difficult to do. Third, we think that HDV and HCV interfere with each other, perhaps in a stronger manner than the previously known HDV/HBV interference (Alfaiate et al., 2016, Lutgehetmann et al., 2012), and this will be the subject of a further study of our team. Indeed, it is possible that direct or indirect (*e.g.*, immune-mediated) interference mechanisms may impede in the long term HDV/HCV co-infections *in vivo*, though they may occur in different contexts. For example, HDV-induced activation of the innate immune response, which is known to have little/no effect on HDV itself (Alfaiate et al., 2016, Zhang et al., 2018), may impact several markers of co-infection, such as HCV that is interferon-sensitive in contrast to HBV (Lutgehetmann et al., 2011, Mutz et al., 2018). Thus, **if this is acceptable, we respectfully request to this Reviewer that we do not display the IF results from the infected mice** as they warrant further studies beyond the scope of this first study. We also believe that the results IF analyses of co-infection *in vitro* requested by this Reviewer are meaningful to establish the co-presence of HDV and live helper viruses.

Finally, using the above-mentioned PCR banding assays, we show that sera from mice co-infected with HDV and HCV or HBV contain RNAs at the expected genomic size of 1.7 kb, in agreement with the presence these RNAs by RTqPCR, which supports our conclusion that the unconventional HDV particles can be secreted and propagated *in vivo*.

5. Finally, the HCV or dengue virus infections in humanized chimera mice took a lot of effort and showed intriguing results. Other than insufficient virological data as mentioned in point 3, the authors may need to study the natural HCV/HDV coinfection in human intravenous drug abusers who frequently co-infected by HBV/HDV/HCV. Do these patients carry HDV RNA within HCV envelope ?

Reply: This is clearly a highly important and interesting question that we wish to pursue in the follow up work of this pioneer study. Indeed, how to best address this and reach statistical significance, given that natural HCV/HDV coinfections in human must be very rare in our opinion, is the subject of on-going discussions with clinicians who may collaborate with us on this issue. Yet, identifying and forming the different patient cohorts (*e.g.*, from human intravenous drug abusers, as proposed by this Reviewer) or just accessing to collections of samples will require time, not only because we do not yet know exactly which are the best types of individuals to screen but also because getting the necessary ethical permits (and funding) is a difficult enterprise, particularly when it deals with countries where HDV is currently prevalent, like Mongolia and South America in the Amazonian basin. Finally, while Western countries, particularly Italia, have been severely hit by HDV infection in the 80's, recovering the collections of specimens from infected patients and their complete clinical description is difficult and will also take several months.

Figure 1. Immunohistochemistry of livers from HDV/HCV co-infected HuHep mice. 4-8 weeks old NOD-FRG mice were engrafted with primary human hepatocytes (PHH). After ca. 2-3 months, the animals displaying HSA levels >15 mg/mL were split in different groups that were infected with HDV alone (10^7 GE/mouse) (**top panels**), with HDV and HBV (10^8 GE/mouse) (**lower left panel**), or with HDV and HCV (1.5×10^5 FFU/mouse) (**lower right panel**). Tissue samples of animals sacrificed at week 14 post-infection (see Figure 7A) were fixed in 10% buffered formalin and embedded in paraffin. 4 μ m-thick tissue sections of formalin-fixed, paraffin-embedded tissue were prepared according to conventional procedures.

For immunofluorescence, sections were incubated with a rabbit anti-HDAg antibody, then the OmniMap Detection Kit was used with a FITC kit (**lefts panels**) or with a rhodamine kit (**right panels**). After stripping of the first antibody, sections were incubated with a human anti-HBcAg with an OmniMap Detection Kit and rhodamine kit (**lower right panel**) or with a mouse anti-NS5A 9E10 with an OmniMap Detection Kit and a rhodamine kit (**lower left panel**). Dapi were used for counterstaining. The slides were scanned with a panoramic scan II (3D histech, Hungary). Scale bars represent 20 μ m.

Note that livers of animals infected with HDV alone did not display HDAg-positive cells, owing to the late time point at sacrifice and probable extinction of HDV RNA replication in the absence of helper virus. In contrast, cells mono-infected by HDV (**green arrows in left panel and red arrows in right panel**), by HBV (**red arrows in lower left panel**) or HCV (**green arrows in lower right panel**), or co-infected could be detected (**yellow arrows in lower panels**). Note that the number of HDV/HCV co-infected cells is significantly lower than for HDV/HBV co-infection, in line with the lower viremia of the former (see Figure 7B and Supplemental Figures 7 & 8), and that HDV/HCV HDAg immunofluorescence is less bright than for the latter, which denotes negative interference between the two viruses.

Literature cited in this rebuttal document

- Alfaiate D, Lucifora J, Abeywickrama-Samarakoon N, Michelet M, Testoni B, Cortay JC, Sureau C, Zoulim F, Deny P, Durantel D (2016) HDV RNA replication is associated with HBV repression and interferon-stimulated genes induction in super-infected hepatocytes. *Antiviral Res* 136: 19-31
- Arai T, Matsumoto K, Saitoh K, Ui M, Ito T, Murakami M, Kanegae Y, Saito I, Cosset FL, Takeuchi Y, Iba H (1998) A new system for stringent, high-titer vesicular stomatitis virus G protein-pseudotyped retrovirus vector induction by introduction of Cre recombinase into stable prepackaging cell lines. *J Virol* 72: 1115-21
- Bauhofer O, Ruggieri A, Schmid B, Schirmacher P, Bartenschlager R (2012) Persistence of HCV in quiescent hepatic cells under conditions of an interferon-induced antiviral response. *Gastroenterology* 143: 429-38 e8
- Bissig KD, Wieland SF, Tran P, Isogawa M, Le TT, Chisari FV, Verma IM (2010) Human liver chimeric mice provide a model for hepatitis B and C virus infection and treatment. *J Clin Invest* 120: 924-30
- Bordier BB, Ohkanda J, Liu P, Lee SY, Salazar FH, Marion PL, Ohashi K, Meuse L, Kay MA, Casey JL, Sebti SM, Hamilton AD, Glenn JS (2003) In vivo antiviral efficacy of prenylation inhibitors against hepatitis delta virus. *J Clin Invest* 112: 407-14
- Calattini S, Fusil F, Mancip J, Dao Thi VL, Granier C, Gadot N, Scoazec JY, Zeisel MB, Baumert TF, Lavillette D, Dreux M, Cosset FL (2015) Functional and Biochemical Characterization of Hepatitis C Virus (HCV) Particles Produced in a Humanized Liver Mouse Model. *J Biol Chem* 290: 23173-87
- Gudima S, He Y, Meier A, Chang J, Chen R, Jarnik M, Nicolas E, Bruss V, Taylor J (2007) Assembly of hepatitis delta virus: particle characterization, including the ability to infect primary human hepatocytes. *J Virol* 81: 3608-17
- Iwamoto M, Watashi K, Tsukuda S, Aly HH, Fukasawa M, Fujimoto A, Suzuki R, Aizaki H, Ito T, Koiwai O, Kusahara H, Wakita T (2014) Evaluation and identification of hepatitis B virus entry inhibitors using HepG2 cells overexpressing a membrane transporter NTCP. *Biochem Biophys Res Commun* 443: 808-13
- Kuo MY, Chao M, Taylor J (1989) Initiation of replication of the human hepatitis delta virus genome from cloned DNA: role of delta antigen. *J Virol* 63: 1945-50
- Li YJ, Macnaughton T, Gao L, Lai MM (2006) RNA-templated replication of hepatitis delta virus: genomic and antigenomic RNAs associate with different nuclear bodies. *J Virol* 80: 6478-86
- Lutgehetmann M, Bornscheuer T, Volz T, Allweiss L, Bockmann JH, Pollok JM, Lohse AW, Petersen J, Dandri M (2011) Hepatitis B virus limits response of human hepatocytes to interferon-alpha in chimeric mice. *Gastroenterology* 140: 2074-83, 2083 e1-2
- Lutgehetmann M, Mancke LV, Volz T, Helbig M, Allweiss L, Bornscheuer T, Pollok JM, Lohse AW, Petersen J, Urban S, Dandri M (2012) Humanized chimeric uPA mouse model for the study of hepatitis B and D virus interactions and preclinical drug evaluation. *Hepatology* 55: 685-94
- Mutz P, Metz P, Lempp FA, Bender S, Qu B, Schoneweis K, Seitz S, Tu T, Restuccia A, Frankish J, Dachert C, Schusser B, Koschny R, Polychronidis G, Schemmer P, Hoffmann K, Baumert TF, Binder M, Urban S, Bartenschlager R (2018) HBV Bypasses the Innate Immune Response and Does Not Protect HCV From Antiviral Activity of Interferon. *Gastroenterology* 154: 1791-1804 e22
- Ni Y, Lempp FA, Mehrle S, Nkongolo S, Kaufman C, Falth M, Stindt J, Koniger C, Nassal M, Kubitz R, Sultmann H, Urban S (2014) Hepatitis B and D viruses exploit sodium taurocholate co-transporting polypeptide for species-specific entry into hepatocytes. *Gastroenterology* 146: 1070-83
- Sainz B, Jr., Chisari FV (2006) Production of infectious hepatitis C virus by well-differentiated, growth-arrested human hepatoma-derived cells. *J Virol* 80: 10253-7
- Watashi K, Urban S, Li W, Wakita T (2014) NTCP and beyond: opening the door to unveil hepatitis B virus entry. *Int J Mol Sci* 15: 2892-905
- Yan H, Zhong G, Xu G, He W, Jing Z, Gao Z, Huang Y, Qi Y, Peng B, Wang H, Fu L, Song M, Chen P, Gao W, Ren B, Sun Y, Cai T, Feng X, Sui J, Li W (2012) Sodium taurocholate cotransporting polypeptide is a functional receptor for human hepatitis B and D virus. *Elife* 1: e00049
- Zhang Z, Filzmayer C, Ni Y, Sultmann H, Mutz P, Hiet MS, Vondran FWR, Bartenschlager R, Urban S (2018) Hepatitis D virus replication is sensed by MDA5 and induces IFN-beta/lambda responses in hepatocytes. *J Hepatol* 69: 25-35

Reviewers' Comments:

Reviewer #1:

Remarks to the Author:

This is a frustrating process. The authors are proposing potentially very interesting and important results. Their overall proposal may well fit very nicely with recent observations that viruses closely related to HDV exist in other species in the absence of any hepadnavirus (Wille, 2018; Hetzel, 2018 - the authors would do well to incorporate this information in their discussion). However, and unfortunately, the manuscript still has problems.

1. Fig. 1D – the image appears to have been manipulated. The right hand side of the HDV, VSV- Δ p and HCV- Δ p lanes has been cropped in the same manner. Furthermore, on close inspection, all three of these “lanes” appear to be identical!!

2. Fig. 1E The immunoprecipitation experiment is still not sufficiently described. How was the elution performed? How did the authors control for non-specific immunoprecipitation/elution? One good way to do this would be to use each of the antibodies against each of the supernatants. Furthermore, it is unimpressive to see that the efficiency of the IP is under 10%. I have taken the liberty to graph the data without the log representation:

3. The EM results are nice, but the authors should purify the particles by immunoprecipitation rather than heparin and should compare particles produced with and without HDV – otherwise, how do we know that the particles observed have anything to do with HDV?

4. There are several problems with the description of the strand-specific assay described on lines 497 – 514. The primers listed are opposite the sense that they should be; that is, the primer 5'-CCCGGCTAC..., is genomic sense and would be appropriate for detecting the antigenome, not the genome. Likewise for the other primer, which for some reason contains two mismatches to the HDV sequence. Moreover, without more information about the RNAs used as standards (presumably they are linear), it is not possible to evaluate the strand-specificity. Mis-priming during the RT step on circular HDV RNA can decrease strand-specificity in ways that would not be detected using linear RNA templates. One might get around this problem by using RNAs of dimer or greater length.

5. The RT-PCR results showing detection of full-length HDV RNA must include controls run without RT. Amplification of the full-length HDV RNA by RT-PCR is difficult and this reviewer is not aware of other reports to have done so. Although the authors certainly may have succeeded in this task, because HDV replication was initiated by transfection of an HDV trimer plasmid DNA, it is possible – even likely – that the plasmid was not completely eliminated by the DNase digestion and that the assay is simply

detecting the HDV plasmid.

6. Is the cytotoxicity really about 40% (Fig. S2A)?? If so, we are all wasting our time.

Reviewer #2:

Remarks to the Author:

The authors have carefully addressed the points that I had raised during the first round of review. Please check the manuscript carefully regarding any references of "expressed HDV", e.g. line 240 "As control, we performed HBV infection assays in Huh-106 cells expressing HDV". This should be corrected throughout.

Congratulations on this very nice, intriguing body of work!

Reviewers' comments:

Reviewer #1 (Remarks to the Author):

This is a frustrating process. The authors are proposing potentially very interesting and important results. Their overall proposal may well fit very nicely with recent observations that viruses closely related to HDV exist in other species in the absence of any hepadnavirus (Wille, 2018; Hetzel, 2018 - the authors would do well to incorporate this information in their discussion). However, and unfortunately, the manuscript still has problems.

Reply: We feel sorry about this comment. The revision of our work and manuscript was very intense and, we believe, allowed incorporation of all the points raised by this Reviewer as well as of most points of the other Reviewers. As for the remaining issues, we have done our best to address them in this new revised manuscript.

We have incorporated in the revised Introduction (page 4) the recent observations that viruses closely related to HDV exist in other species in the absence of any hepadnavirus, which indeed gives strong credit to our observations that HDV can be transmitted by hepadnavirus-unrelated viruses.

1. Fig. 1D – the image appears to have been manipulated. The right hand side of the HDV, VSV- Δ p and HCV- Δ p lanes has been cropped in the same manner. Furthermore, on close inspection, all three of these “lanes” appear to be identical!!

Reply: We are humbly asking to accept our most sincere apologies for not having detected the problem with this image, which obviously shows that the three lanes (HDV, VSV- Δ P, HCV- Δ P in our Fig 1D) are absolutely identical. After careful examination of the image, it appears that the 4 lanes (*i.e.*, with the positive control (*Ctrl* in Fig 1D)) are the same. We are sorry to say that we have been too quick, owing to the intensity of the revision work and we should have been able to immediately detect this absurd occurrence. In addition, we are most grateful with this Reviewer for detecting the mistake.

We truly do not understand what happened. The only bit of explanation that we think about would be a bizarre event generated with a Bio-Rad ChemiDoc™ Imager that seems to have replicated part of the image (likely part of the *Ctrl* lane). I must say that I have never come across such a situation during my career. Actually, we usually use in most of our work (including this study) the Odyssey Imaging System from LI-COR Biosciences as it allows linear integration of signals on scales of several logs which is essential when dealing with quantifications (see, for example, the Western blot image of Fig 3E that was generated with this system). However, because of a breakdown of this equipment that occurred during the revision period and of its subsequent repairing time, we had to use a Bio-Rad ChemiDoc™ Imager for the image of Fig 1D. We thus believe that either we did not use it properly or, alternatively, that something got wrong in the system, but the situation is this one: we were happy when we saw the three pairs of bands corresponding to the exact sizes of L-HDAg and S-HDAg, and we did not - and I apologize for this - think further than this.

We feel deeply sorry with the confusion it may have generated, or worse, with the possible opinion of this Reviewer who may think that the image would have been manipulated. I am respectfully asking to believe that it is certainly not the case and I trust that my professional reputation based on over 30 years of academic research in molecular virology confirms my intellectual honesty and that of my team colleagues.

Meanwhile, we have redone this Western blot using the original samples (and processed them with our Odyssey Imaging System), which is now displayed in the new revised Figure 1D.

Once again, please accept our apologies and, above everything, please be convinced of our scientific integrity.

2. Fig. 1E The immunoprecipitation experiment is still not sufficiently described. How was the elution performed? How did the authors control for non-specific immunoprecipitation/elution? One good way to do this would be to use each of the antibodies against each of the supernatants. Furthermore, it is unimpressive to see that the efficiency of the IP is under 10%. I have taken the liberty to graph the data without the log representation:

Reply: We provide a better description of the method used for this immunoprecipitation experiment in our revised manuscript (page 16). To address the concerns of this Reviewer, we now provide the results of immunoprecipitation experiments that were controlled as per his/her request, *i.e.*, using each of the antibodies against each of the supernatants (new Figure 1e). As also suggested by this Reviewer, we now display the data without the log representation. We have also worked to optimize the

antigen/antibody ratio in order to improve the recovery of HDV particles (now, of over 15%). Yet, we propose that the relatively low efficiency of this immunoprecipitation reflects the strong competition by HBV, HCV and VSV subviral particles that outnumber the HDV RNA-bearing particles.

3. The EM results are nice, but the authors should purify the particles by immunoprecipitation rather than heparin and should compare particles produced with and without HDV – otherwise, how do we know that the particles observed have anything to do with HDV?

Reply: We thank this Reviewer for his/her positive appreciation of these EM results that were performed as part of our reply to Comment #3 of Reviewer #3 who suggested to provide “a simple EM study (...) to visualize the size, distribution of these pseudo-type HDV particles”. Rather, owing to the presence of contaminants detected *via* simple EM studies (data not shown), we undertook a more sophisticated EM study of particles purified with heparin beads (Figure 1F and Supplemental Figure 3) with the help of the EM facility on our campus.

We respectfully request to this Reviewer that we can keep the EM results as they are in this report for the following reasons. Indeed, the purification of the particles by immunoprecipitation with viral surface GP antibodies may potentially improve the analysis but would not allow distinguishing HDV particles from the subviral particles of either helper virus as they both share the same surface antigen. Furthermore, we are concerned that such a study for which we would have to determine and optimize many conditions for 3 different viruses investigated simultaneously (HDV, HCV- Δ P, VSV- Δ P) would require a tremendous effort, of largely over a year, and would not provide incremental results relative to the set of data in this report. Particularly, from our own experience, eluting antibody-captured particles from beads or equivalent material is technically a very difficult issue and also depends on variable antigen/antibody affinities between the three virus types. Given the uncertainty of these outcomes, we think that this is beyond what is reasonably needed to characterize these novel particles for which we believe we have provided multiple and convergent evidence with different biochemical and functional *in vitro* or *in vivo* assays in this report. Finally, that we did not detect particles when pSVLD3 was expressed alone (Figure 1F), that we detected HDV particles in addition to subviral particles when pSVLD3 and GP-expression plasmids were co-expressed (Figure 1F and Supplemental Figure 3) but that we only detected subviral particles when GP-expression plasmids were individually expressed (Supplemental Figure 3) argues that the particles observed are HDV particles.

4. There are several problems with the description of the strand-specific assay described on lines 497 – 514.

Reply: We thank this Reviewer for inviting us to clarify our methodologies that have been modified in the Methods section of our new revised manuscript. The strand-specific RTqPCR method utilized in the present study to discriminate and quantify genomic and antigenomic HDV RNA consists in reverse transcription reactions with strand-specific primers (opposite polarity as compared to the template) followed by qPCR reaction. We address below the five concerns of his/her comment.

Concern #1: *The primers listed are opposite the sense that they should be; that is, the primer 5'-CCCGGCTAC....., is genomic sense and would be appropriate for detecting the antigenome, not the genome.*

Reply: We apologize for our sentence “... RNAs were reverse transcribed ... using strand-specific oligonucleotides primers for genomic RNA 5'-CCGGCTACTTCTTTCCCTTCTCTCGTC and for antigenomic RNA 5'-CACCGAAGAAGGAAGGCCCTGGAGAACAA” that was unclear. It has been replaced by “..., extracted RNAs were reverse transcribed ... by using specific primer: genomic primer 5'-CCGGCTACTTCTTTCCCTTCTCTCGTC for genomic-sense cDNA synthesis and antigenomic primer 5'-CACCGAAGAAGGAAGGCCCTGGAGAACAA for antigenomic-sense cDNA synthesis” in the revised Methods (page 15).

Concern #2: *Likewise for the other primer, which for some reason contains two mismatches to the HDV sequence.*

Reply: For this primer (5'-CACCGAAGAAGGAAGGCCCTGGAGAACAA), which detects the genomic HDV RNA and allows amplification of antigenomic-sense cDNA, its two mismatches to the HDV sequence in Genbank M21012 are due to the fact that when we sequenced the pSVLD3 plasmid from the EBV-Rev primer (1551-1570 on pSVLD3 Addgene sequence), we detected these mismatches at positions 1754 and 1768 in the pSVLD3 Addgene sequence. We therefore decided to synthesize this primer in order to best match our sequence and also because the identical primer was used in a previous description of the HDV RNA strand-specific RTqPCR assay by Li et al., (2006).

Concern #3: Moreover, without more information about the RNAs used as standards (presumably they are linear), it is not possible to evaluate the strand-specificity.

Reply: We have improved in our revised Methods section the description of the genomic and antigenomic HDV RNAs used as standards for this strand-specific RTqPCR assay (page 15). They were obtained by *in vitro* transcription of a set of HDV DNA amplicons flanked by T7 promoters using a commercially available kit according to manufacturer's instructions (RiboMAXTMExpress T7, Promega). The full-length linear RNAs were treated with RNase-free-DNase, followed by RNA purification (GeneJET RNA purification kit, Thermo Fisher Scientific) and quantified using a Nanodrop device (Thermo Fisher Scientific). The genomic or antigenomic HDV RNA calibration standards were diluted in RNase-free water and stored at -80°C in single use aliquots. The characterization of genomic and antigenomic RNA standards (artificial RNA) with the correct primer and the respective opposite primer were performed as described previously by Giersch et al. 2017; Freitas et al 2012; Gudima et al 2007.

Concern #4: Mis-priming during the RT step on circular HDV RNA can decrease strand-specificity in ways that would not be detected using linear RNA templates.

Reply: Mis-priming is defined as non-specific DNA-primer binding and extension during reverse transcription. Regarding HDV genomic and antigenomic HDV RNA, since both molecules display high degree of intramolecular base pairing it must be considered two types of mispriming: within the same molecule (intramolecular mis-priming) or within a molecule of reverse polarity (intermolecular mis-priming). Strand specific cDNA synthesis was carried out using specific primers (to minimize RT priming at multiple points as with random primers) using a reverse transcriptase with impaired/reduced ribonuclease-H activity under a significantly low temperature incubation, two factors shown to reduce significantly the potential of template switch events during reverse transcription. Using RNaseH⁻ RTs, during elongation of the nascent DNA chain, the RNA template is not degraded and remains as a double stranded DNA/RNA molecule of high stability disfavoring the dissociation of the RT from the template. In other words, as the nascent cDNA remains complexed to the RNA template, it is unlikely to find and bind a different acceptor RNA to allow template switching and production of chimeric cDNAs or cDNAs carrying internal deletions.

The possibility of intermolecular mis-priming was experimentally checked and the results obtained, as presented in the Supplemental Figure 1F indicate that intermolecular mis-priming is a very rare event. Specifically, it is as low as 0.000001% using 10¹¹ copies of genomic RNA as template and a primer of the same polarity and even lower when reverse transcription was carried out using 10¹¹ copies of antigenomic HDV RNA as template and a primer of antigenomic polarity, suggesting that in our experimental conditions, the primer of genomic polarity almost exclusively detects antigenomic HDV RNA and *vice-versa*.

Intramolecular mispriming, defined as non-specific binding of the DNA-primer to the same molecule was evaluated by assessing the linearity of the reverse transcription step. We foresee that mispriming upstream of the specific primer-binding site would not interfere with specific primer binding and extension (cDNA synthesis) and quantification by qPCR. The reverse scenario, mispriming downstream the specific primer-binding site could have a more important impact, although almost impossible to predict, on the kinetics of primer extension and qPCR detection sensitivity. To determine the efficiency of RNA-to-cDNA conversion we analyzed the linearity and range of the designed RTqPCR assay. We performed cDNA synthesis using a 10-fold serial dilution ranging from 10¹¹ to 10 copies of the *in vitro* synthesized genomic and antigenomic HDV RNA and the results show an efficient and RNA dose-dependent cDNA synthesis that results in a qPCR standard calibration curve with a slope of -3.3 ± 10% reflecting an efficiency close to 100%. It is our opinion that the linearity of the RT reaction over this wide range suggests rare intramolecular mispriming events. We also carefully analyzed the dissociation curves of the products obtained from qPCR and there is no indication of any extra subproduct being produced other than the expected correct amplicon of less than 100bps with a single dissociation temperature.

Concern #5: One might get around this problem by using RNAs of dimer or greater length.

Reply: Unit length linear HDV RNA standards should closely mimic the circular HDV RNA, the only type of HDV RNA molecules incorporated into particles that are secreted from the cells. The major difference is that HDV genome inside the particles has no end. Our data strongly suggests that our strand specific genomic and antigenomic HDV RNA quantification by RT-qPCR is not significantly biased by inter or intramolecular mispriming. We anticipate that the number of mispriming events between a linear and a circular RNA molecule of the same size would be similar. Therefore, only one concern remains, strand displacement during cDNA synthesis resulting in HDV cDNA concatemers that would lead to an over-estimation of the input RNA by qPCR. Because, RTs have limited processivity (Mohr et al., 2013), this consideration is likely to have little effect on large circRNAs (Szabo and Salzman, 2016). For the reasons

explained above, it is our opinion that there would be no advantage or significant improvement in our assays from using dimeric or greater length RNA standards.

Finally, we wish to recall that there is not a standard protocol for quantification of HDV RNA by RTqPCR though several groups regularly use one-step RT-qPCR assays (e.g., Ferns et al., 2012; Scholtes et al., 2012; Karatayli et al., 2014). Likewise for calibration standards, there are no recognized calibration standards available for HDV RNA quantification and the quantitative data generated by different laboratories are produced by different RTqPCR protocols that use HDV plasmid DNA (Le al., et al 2005; Hofmann et al., 2010; Mederacke et al., 2010; Zachou et al., 2010) or *in vitro*-transcribed genomic or antigenomic HDV RNA standards (Freitas et al., 2012; Gudima et al., 2007; Giersch et al., 2017) for calibration of assays. Interestingly, despite their differences, these protocols give comparable results for the HDV RT-qPCR assays.

5. The RT-PCR results showing detection of full-length HDV RNA must include controls run without RT. Amplification of the full-length HDV RNA by RT-PCR is difficult and this reviewer is not aware of other reports to have done so. Although the authors certainly may have succeeded in this task, because HDV replication was initiated by transfection of an HDV trimer plasmid DNA, it is possible – even likely – that the plasmid was not completely eliminated by the DNase digestion and that the assay is simply detecting the HDV plasmid.

Reply: We provide in the revised Figure 1B the controls run without RT and/or without DNase treatment. These data indicate that we could detect full-length HDV RNA from replicated RNAs but not from HDV trimer plasmid DNA. Please note that the “No GP” control, which does not yield full-length HDV RNA detection in the transfected cells supernatants, as well as the detection of full-length HDV RNA in the gradient (Figure 1G), in the supernatants of co-infected cells (Figure 6B, F & J) or in the sera of co-infected animals (Supplemental Figure 7B) also indicate that this method detects replicated and secreted HDV RNA. Furthermore, no PCR signal could be detected in the supernatants using a genomic primer during the reverse transcription step before PCR amplification, indicating absence of antigenomic HDV RNA secretion

6. Is the cytotoxicity really about 40% (Fig. S2A)?? If so, we are all wasting our time.

Reply: We thank this Reviewer for pointing out this mistake in the display of this Fig. S2A. Indeed, to experimentally address the Point #3 of this Reviewer and the Point #2 of Reviewer #2 (“2. In figure 1a, the release of virus through cell death should be ruled out.”) of our previous revision, we had tested several variations from our HDV production protocol, including production in media with or without 2% DMSO, OptiMEM, DMEM+FBS (different FBS doses) and daily change of media vs. 6 day-long conditioned media. However, by error, the previous Fig. S2A showed the toxicity results from supernatants of cells transfected with the indicated constructs and cultivated using the latter condition, *i.e.*, harvested at day 6 post-transfection without daily changing of the media, which we generally do not use in our experiments and which raises cytotoxicity levels.

In our standard conditions (adapted from Ref #29), *i.e.*, for which transfected cells are maintained in a daily-changed William’s E medium supplemented with 2% FBS +/- 2% DMSO, the average cytotoxicity at day 6 is below 15%, as shown in the revised Fig. S2A, and is similar to that of non-transfected cell cultured in parallel. Note that this mistake only concerns Fig. S2A but not the other panels. We have corrected this figure in the revised manuscript and we apologize for the inaccuracy.

Reviewer #2 (Remarks to the Author):

The authors have carefully addressed the points that I had raised during the first round of review. Please check the manuscript carefully regarding any references of “expressed HDV”, e.g. line 240 “As control, we performed HBV infection assays in Huh-106 cells expressing HDV”. This should be corrected throughout.

Reply: We thank this Reviewer for his/her positive appreciation of our revised manuscript and for pointing out these sentences that are corrected throughout in the new revised manuscript.

Congratulations on this very nice, intriguing body of work!

Reply: We thank this Reviewer for his/her nice comment on our study!

Reviewers' Comments:

Reviewer #1:

Remarks to the Author:

The authors have adequately addressed the concerns raised in the prior reviews. The authors' findings are very interesting and important for the HDV field, and, as I mentioned in the previous review, provide an intriguing fit with recent reports (some published), based on metagenomic analysis, that viruses very closely related to HDV exist in other species in the absence of any detectable hepadnavirus.

Reviewers' Comments:**Reviewer #1 (Remarks to the Author):**

The authors have adequately addressed the concerns raised in the prior reviews. The authors' findings are very interesting and important for the HDV field, and, as I mentioned in the previous review, provide an intriguing fit with recent reports (some published), based on metagenomic analysis, that viruses very closely related to HDV exist in other species in the absence of any detectable hepadnavirus.

Reply: We thank this Reviewer for his/her positive appreciation of our revised manuscript and comment on our study.